# Screening an In-House Isoquinoline Alkaloids Library for New Blockers of Voltage-Gated Na^+^ Channels Using Voltage Sensor Fluorescent Probes: Hits and Biases

**DOI:** 10.3390/molecules27134133

**Published:** 2022-06-28

**Authors:** Quentin Coquerel, Claire Legendre, Jacinthe Frangieh, Stephan De Waard, Jérôme Montnach, Leos Cmarko, Joseph Khoury, Charifat Said Hassane, Dimitri Bréard, Benjamin Siegler, Ziad Fajloun, Harold De Pomyers, Kamel Mabrouk, Norbert Weiss, Daniel Henrion, Pascal Richomme, César Mattei, Michel De Waard, Anne-Marie Le Ray, Christian Legros

**Affiliations:** 1Univ Angers, INSERM, CNRS, MITOVASC, Equipe CarME, SFR ICAT, 49000 Angers, France; quentinco@fastmail.com (Q.C.); claire.legendre@univ-angers.fr (C.L.); jacynthefrangieh@gmail.com (J.F.); joseph-0-khoury@hotmail.com (J.K.); charifat.said-hassane@univ-reunion.fr (C.S.H.); daniel.henrion@univ-angers.fr (D.H.); cesar.mattei@univ-angers.fr (C.M.); 2Nantes Université, CHU Nantes, CNRS, INSERM, L’institut du Thorax, 44000 Nantes, France; stephan.dewaard@univ-nantes.fr (S.D.W.); jerome.montnach@univ-nantes.fr (J.M.); leos.cmarko@univ-nantes.fr (L.C.); michel.dewaard@univ-nantes.fr (M.D.W.); 3Institute of Biology and Medical Genetics, First Faculty of Medicine, Charles University, 128 00 Prague, Czech Republic; 4Institute of Organic Chemistry and Biochemistry of the Czech Academy of Sciences, 166 10 Prague, Czech Republic; 5Laboratory of Applied Biotechnology, AZM Centre for Research in Biotechnology and Its Application, Doctoral School for Sciences and Technology, Tripoli 1352, Lebanon; zfajloun@gmail.com; 6Univ Angers, SONAS, SFR QUASAV, 49100 Angers, France; dimitri.breard@univ-angers.fr (D.B.); benjamin.siegler@univ-angers.fr (B.S.); pascal.richomme@univ-angers.fr (P.R.); 7Latoxan Laboratory, 26800 Portes lès Valence, France; harold.pomyers@latoxan.com; 8CNRS, ICR, Aix Marseille Univ, 13397 Marseille, France; kamel.mabrouk@univ-amu.fr; 9Department of Pathophysiology, First Faculty of Medicine, Charles University, 160 00 Prague, Czech Republic; nalweiss@gmail.com

**Keywords:** voltage-gated sodium channel, isoquinoline alkaloids, oxoaporphine, voltage sensor probes, Na^+^ fluorescent probe ANG-2, GH3b6 cells

## Abstract

Voltage-gated Na^+^ (Na_V_) channels are significant therapeutic targets for the treatment of cardiac and neurological disorders, thus promoting the search for novel Na_V_ channel ligands. With the objective of discovering new blockers of Na_V_ channel ligands, we screened an In-House vegetal alkaloid library using fluorescence cell-based assays. We screened 62 isoquinoline alkaloids (IA) for their ability to decrease the FRET signal of voltage sensor probes (VSP), which were induced by the activation of Na_V_ channels with batrachotoxin (BTX) in GH3b6 cells. This led to the selection of five IA: liriodenine, oxostephanine, thalmiculine, protopine, and bebeerine, inhibiting the BTX-induced VSP signal with micromolar IC_50_. These five alkaloids were then assayed using the Na^+^ fluorescent probe ANG-2 and the patch-clamp technique. Only oxostephanine and liriodenine were able to inhibit the BTX-induced ANG-2 signal in HEK293-hNa_V_1.3 cells. Indeed, liriodenine and oxostephanine decreased the effects of BTX on Na^+^ currents elicited by the hNa_V_1.3 channel, suggesting that conformation change induced by BTX binding could induce a bias in fluorescent assays. However, among the five IA selected in the VSP assay, only bebeerine exhibited strong inhibitory effects against Na^+^ currents elicited by the hNav1.2 and hNav1.6 channels, with IC_50_ values below 10 µM. So far, bebeerine is the first BBIQ to have been reported to block Na_V_ channels, with promising therapeutical applications.

## 1. Introduction

Voltage-gated sodium channels (Na_V_ channels) are heteromultimeric membrane proteins that play a major role in neuronal excitability and the excitation–contraction coupling of myocytes [1]. They are composed of a large pore-forming α-subunit encoded by nine genes (*scn1a*, *2a*, *3a*, *4a*, *5a*, *8a*, *9a*, *10a,* and *11a*), thus giving a total of nine Na_V_ channel subtypes (Na_V_1.1–Na_V_1.9) [2]. Mutations leading to the gain of function of Na_V_ channels are responsible for a wide variety of pathologies, including epilepsy, chronic pain, and cardiac arrhythmia [3]. Therefore, searching for new Na_V_ channel modulatory drugs remain highly attractive for clinical and pharmaceutical purposes. In addition, Na_V_ channels represent a therapeutically validated target for five categories of inhibitory drugs: (i) anti-arrhythmics, (ii) anti-convulsants, (iii) antiepileptics, (iv) anesthetics, and (v) analgesics [4].

Alkaloids constitute a large and heterogeneous group of basic, heterocyclic, nitrogen-containing, natural compounds with a tremendous number of chemical structures [5]. They are mainly produced by plants, but they can also be found in bacteria, fungi, and animals. Although they are generally toxic, they also exhibit therapeutical properties with multi-target activities [6,7]. More than 27,000 structures of alkaloids have been described, and several have been used in traditional and conventional medicine, such as in analgesic (codeine, morphine), anti-tumoral (taxol), anti-arrhythmic (quinidine), antispasmodic (atropine, hyoscyamine), muscle relaxant (tubocurarine), and antihypertensive (canescine, rescinnamine) drugs [6,7]. Despite having been studied for decades, the high structural diversity of alkaloids with regard to their pharmacological properties makes them a highly relevant chemical group for drug screening assays.

Eight distinct Na_V_ channel binding sites (1 to 8) have been described for natural ligands [4,8]. Historically, tetrodotoxin (TTX) and saxitoxin (STX), two hydrophilic microbial neurotoxins that are structurally related to guanidine alkaloids, were the first to be characterized as potent inhibitors of Na_V_ channels through their binding to Site 1, localized on the external side of the ion channel pore [4,9,10]. Since then, Na_V_ channel subtypes have been classified as TTX-sensitive (TTX-S) or TTX-resistant (TTX-R) because they are blocked by nanomolar and micromolar concentrations, respectively [11,12].

Other alkaloids with various structures, mainly of vegetal origin, target Na_V_ channels (Figure 1). Lipid-soluble alkaloid toxins, including batrachotoxin (BTX) and veratridine (VTD) as well as aconitine bind to Site 2 [4,8]. These toxins bind to open Na_V_ channels and maintain them in this configuration, thus preventing their transition to the inactivation state [8,13,14]. BTX, VTD, and aconitine are considered as Na_V_ channel activators, which lead to the membrane depolarization of excitable cells and their firing [8]. BTX binds to Na_V_ channels with very high affinity, behaving as a full activator, whereas VTD and aconitine behave as partial activators [8].

Isoquinoline alkaloids (IA) represent one of the most diverse and abundant groups of plant compounds. They are biosynthesized from tyrosine, leading to several chemical subgroups such as benzylisoquinoline, aporphine, protoberberine, protopine, morphinan, and many others [7,26]. Until now, only two of the large aporphine alkaloid subgroups (>1000 members) have been described as Na_V_ channel blockers: liriodenine from *Fissistigma glaucescens* [27] and crebanine from *Stephania* sp. [28] (Figure 1). Both compounds exhibit anti-arrhythmic activities mediated by the inhibition of multiple cardiac ion channels, such as the Na_V_, Ca_V,_ and K_V_ channels [27,28,29]. However, this IA group has been poorly studied for their effects on Na_V_ channels. Therefore, in this work, we aimed to identify new Na_v_ channel blockers among IA from an in-house library composed of plant alkaloids.

We focused our studies on the IA subgroups aporphine, 1-benzyl isoquinoline (BIQ), protoberberine, and bisbenzylisoquinoline (BBIQ) in our search for new blockers of Na_V_ channels. The screening assay consists in the measurement of the FRET voltage sensor probe (VSP) signal elicited by the activation of Na_V_ channels that are endogenously expressed in GH3b6 cells by BTX in the absence and in the presence of 62 IA from our in-house natural compounds library. This allowed us to identify five putative Na_V_ channel blockers: liriodenine, oxostephanine, thalmiculine, protopine, and bebeerine. These five IA were assayed for their ability to block the ANG-2 fluorescence emission induced by BTX in HEK293-expressing hNav1.3 and then to inhibit Na^+^ currents elicited by the hNav1.2, hNav1.3, and hNav1.6 channels via automated patch-clamp electrophysiology. Our findings highlighted that the use of the activators of Na_V_ channels, such as BTX, which exhibits a “foot-in-the-door” mechanism and thus freezes the Na_V_ channels in an open state, could compromise the discovery of selective and potent inhibitors.

## 2. Results

### 2.1. VSP-FRET Assays Using GH3b6 for Pharmacological Characterization of Na_V_ Channels

The GH3b6 cell line was selected for its wide use as a cellular model for pharmacological studies on Na_V_ channels [30,31,32,33,34,35]. Here, we used RT-qPCR to establish the Na_V_ channel gene expression profile of GH3b6 cells. Our data showed the significant expression of four Nav channel transcripts (*scn1a*, *scn2a*, *scn3a,* and *scn8a*), which encoded four neuronal and TTX-sensitive Na_V_ channel subtypes (Na_V_1.1, 1.2, 1.3, and 1.6, respectively) (Figure 2). The rank order of the expression levels of these transcripts was *scn3a* > *scn8a* > *scn2a* > *scn1a* (*p* < 0.0001). The amplification of *scn3a* was observed with the lowest Ct-values (Ct = 22.9, for 10 ng of ADNc). Our data indicate that GH3b6 cells mainly expressed the neuronal TTX-sensitive Na_V_1.3 channel subtype, which thereby tends to be consistent with earlier observations [35,36,37].

Next, we measured the fluorescent resonance transfer (FRET) between CC2-DMPE and DisBAC2(3) in GH3b6 cells to characterize the change in the membrane potential induced by BTX as it is considered as a full activator of Na_V_ channels [8,13,14]. The fluorescence emission of CC2-DMPE increased, which is in parallel with the decrease of DisBAC2(3) fluorescence in the presence of increasing concentrations of BTX (from 0.01 to 3 µM) (Figure 3a). Subsequently, the VSP fluorescence emission ratio increased in a concentration-dependent manner. The data were best fitted by the Hill–Langmuir equation with a variable slope for a bimolecular reaction with an EC_50_ value of 0.27 ± 0.06 µM (Figure 3b). At 100 nM, TTX fully abolished the VSP fluorescence emission ratio induced by 1 µM BTX (Figure 3b). The TTX inhibitory effects were fitted as described above, with an IC_50_ value of 8.58 ± 1.67 nM (Figure 3b). Indeed, BTX binds to the open state of Na_V_ channels and consequently prevents their closing in an irreversible manner. Then, the injection of Na^+^, which depolarized the membrane, led to the activation of the Na_V_ channels and allowed for BTX binding. Since this interaction is irreversible, the cells became leaky to Na^+^. This led to an Na^+^ influx, which was counteracted by an Na^+^ efflux mediated by Na^+^/K^+^ ATPase and other Na^+^ transporters (e.g., NCX). This explains the slow kinetic of the fluorescence emission, which reached a plateau. This latter corresponds to an equilibrium between the Na^+^ influx and the Na^+^ efflux. Thus, the VSP signal induced by BTX was inhibited by TTX at a nanomolar concentration, in accordance with the presence of TTX-sensitive Nav channels. Altogether, these results demonstrate that GH3b6 cells are suitable for a FRET-based VSP assay for the screening of Na_V_ channel inhibitors.

### 2.2. Screening Vegetal Alkaloids for Blockers of Na_V_ Channels Using VSP in GH3b6 Cells

Sixty-two IA (from IA1 to IA77) from our in-house library were chosen for their structure, none of which, except for liriodenine, had been described as a Na_V_ channel ligand (see Appendix A, for trivial name, chemical class (CAS number), structure, plant origin, and reference). Thus, we first blind-tested these 62 IA for their ability to inhibit a BTX-induced VSP signal in GH3b6 cells (Figure 4). Figure 4a shows the BTX-induced VSP signal in the absence and in the presence of each IA. Thus, thirty samples exhibited an inhibition rate of 50% and more (Figure 4b). To select the potent inhibitors of Nav channels, we used a threshold of 75% signal inhibition since the IA had been screened at rather high concentrations (10–50 µg/mL, corresponding to 12–20 µM) (Figure 4b). Then, ten IA were selected for further chemical analysis: (1) HPLC monitoring at two different wavelengths (210 nm and 280 nm) to evaluate their purity in %, (2) NMR structure validation, and (3) if necessary, mass spectrometry (MS) analysis and/or optical rotation to confirm IA identity (Table 1). It was only at this stage that the different samples were identified and the blind evaluation was stopped. Samples IA50 (tetrandrine), IA31 (tiliacorine), IA36 (sukhodianine), and IA42 (gyrocarpusine) were rejected because of their low purity (<75% at 210 nm and 280 nm). Therefore, we retained the following five IA for further pharmacological characterization: oxostephanine (IA14 and IA24), liriodenine (IA39), thalmiculine (IA49), bebeerine (IA52), and protopine (IA69) (Table 1).

The NMR spectra of IA24, 14, 39, 49, 52, and 69 confirmed their structures (Appendix A). Sometimes, MS and ${\left[\alpha \right]}_{D}^{20{}^{\circ}C}$ were necessary. Interestingly, oxostephanine and liriodenine are two members of the small subgroup of oxoaporphines.

Oxostephanine, liriodenine, thalmiculine, bebeerine, and protopine inhibited the BTX-induced VSP signal in GH3b6 cells in a concentration-dependent manner, with IC_50_ values in the micromolar range (Figure 5). Oxostephanine and liriodenine were able to abolish the BTX-induced VSP signal with similar IC_50_ values (5–6 µM). In contrast, thalmiculine, bebeerine, and protopine decreased BTX response up to 31–65%, but with lower IC_50_ values (0.4–4 µM) (Table 2). The Hill coefficient values were close to 1 for these five IA, suggesting a bimolecular interaction between them and their molecular targets.

### 2.3. Effects of Oxostephanine, Liriodenine, Thalmiculine, Bebeerine, and Protopine on Intracellular Na^+^ Concentration in a Stable Cell Line Expressing hNa_V_1.3

To further characterize the inhibitory effects of liriodenine, oxostephanine, thalmiculine, protopine, and bebeerine on Nav channels, these IA were assayed on a clonal HEK293 cell line expressing the human Na_V_1.3 channel subtype (HEK293-hNa_V_1.3) (Figure 6). We used the ANG-2 Na^+^ probe, which has been proven to be the most reliable in monitoring the Na^+^ intracellular concentration ([Na^+^]_i_) [38]. At 3 µM, BTX induced a slow increase of the ANG-2 fluorescence ratio, which was totally blocked by 1 µM TTX (92 ± 5%, *p* < 0.0001, Figure 6a,b). Oxostephanine, liriodenine, and bebeerine inhibited the BTX-induced [Na^+^]_i_ elevation in HEK293-hNa_V_1.3 up to 43 ± 12% (*p* < 0.01), 50 ± 10% (*p* < 0.001), and 18 ± 2% (*p* < 0.0001) (Figure 6a,b). However, no significant inhibitory effects on hNa_V_1.3 channels were observed with thamiculine (*p* = 0.08) and protopine (*p* = 0.15) (Figure 6b). Altogether, these results partially agreed with data obtained in VSP assays, showing that only oxostephanine, liriodenine, and bebeerine could be considered as putative inhibitors of Na_V_ channels.

### 2.4. Characterization of the Effects of Oxostephanine, Liriodenine, Thalmiculine, Bebeerine, and Protopine on Na^+^ Currents Elicited by the hNa_V_1.2, hNa_V_1.3, and hNa_V_1.6 Channels

To validate the inhibitory effects of oxostephanine, liriodenine, thalmiculine, bebeerine, and protopine on Na_V_ channels, automated patch-clamp experiments were performed using three different cell lines stably expressing the Na_V_ channel subtypes detected by RT-qPCR in GH3b6 cells. These five IA were assayed on the Na^+^ currents elicited by the depolarizing step in the three cell lines, which stably expressed the hNa_V_1.2 (HEK293- hNa_V_1.2 cells), hNa_V_1.3 (HEK293- hNa_V_1.3 cells), and hNa_V_1.6 channels (HEK293- hNa_V_1.6 cells). At 10 µM, oxostephanine, liriodenine, thalmiculine, and protopine had no effects on the hNa_V_1.2, hNa_V_1.3, and hNa_V_1.6 channels (Figure 7a and Appendix A). At the same concentration, bebeerine was able to inhibit the Na^+^ currents elicited by the hNa_V_1.2, hNa_V_1.3, and hNa_V_1.6 channels (Figure 7a,b). The inhibition rate of bebeerine on these three Na_V_ channels did not depend on the Vm (two-way ANOVA test followed by Holm–Sidak’s multiple comparison test, *p* < 0.001). At a Vm between −20 mV and 30 mV, bebeerine had stronger inhibitory effects on the hNa_V_1.3 (60–64%, *n* = 38) and hNa_V_1.6 channels (70–74%, n = 32) than on the hNa_V_1.2 channel (42–50%, n = 38, Figure 7c). Notably, bebeerine significantly inhibited the hNa_V_1.6 channel to a higher extent (74 ± 3%, *n* = 32) than the hNa_V_1.3 channel (64 ± 4%, *n* = 48; *p* < 0.05), at −20 mV and −15 mV, respectively (Figure 7c). Altogether, these data show that bebeerine efficiently blocked the hNa_V_1.2, hNa_V_1.3, and hNa_V_1.6 channels, with the following rank order: hNa_V_1.6 > hNa_V_1.3 > hNa_V_1.2.

Figure 8 illustrates the effects of bebeerine (at 10 µM) on the voltage-dependent activation and inactivation of the hNa_V_1.2, hNa_V_1.3, and hNa_V_1.6 channels. All fitting parameters are summarized in Table 3. There was no significant change in both the half-activation and inactivation potential (V_1/2_) of the Na^+^ currents elicited by the hNa_V_1.2 channels (*p* = 0.72 and *p* = 0.69, respectively) nor by the hNa_V_1.6 channels (*p* = 0.71 and *p* = 0.32, respectively) in the presence of bebeerine (Figure 8a,c). Concerning the hNa_V_1.3 channel, bebeerine induced a slight negative shift in V_1/2_ activation (−3.6 ± 1.0, *p* < 0.001), but not in V_1/2_ activation (*p* = 0.08, Figure 8b). However, a significant increase of slope values for both the activation and inactivation curves was observed for each Na_V_ channel subtype in the presence of bebeerine (Table 3).

Oxostephanine and liriodenine did not exhibit significant inhibitory effects at 10 µM on Na^+^ currents elicited by the Na_V_1.3 channels, while both IA were able to efficiently inhibit the BTX-induced ANG-2 fluorescence signals in HEK293-hNa_V_1.3 cells. Thus, we assayed these two IA on the Na^+^ currents elicited by the hNa_V_1.3 channel in the presence of 1 µM BTX. As expected, BTX profoundly altered the Na^+^ currents elicited by the hNa_V_1.3 channel by increasing the current densities and inhibition of inactivation (Figure 9a). Interestingly, in the presence of BTX, oxostephanine and liriodenine significantly decreased the current densities of the hNa_V_1.3 channel at −60 mV (Figure 9a–c); however, only IA39 positively shifted the V_1/2_ activation in the presence of BTX, but to a small extent (*p* < 0.01, Figure 9b).

We also examined the effects of oxostephanine and liriodenine on the inactivation process of hNa_V_1.3 in the presence and in the absence of BTX. We evaluated the AUC of currents traces, which illustrate the inactivation rate of hNa_V_1.3 channels. BTX induced the 55-fold increase of the AUC, indicating the decrease of the inactivation kinetic (*p* < 0.0001, data not shown). Interestingly, the slow-down of inactivation induced by BTX was significantly decreased by both IA14 and IA39 (*p* < 0.05, Figure 9b,c).

## 3. Discussion

Our study aimed to identify new ligands of the Nav channels by screening vegetal IA from an in-house library. The IA family offers a large molecular diversity that has not been extensively studied for their activities against Na_V_ channels. Using fluorescent VSP, we identified two oxoaporphine alkaloids (oxostephanine and liriodenine), two BBIQ (bebeerine and thalmiculine), and a protopine (protopine) which inhibited BTX-induced depolarization in GH3b6 cells, with IC_50_ values in the micromolar range. However, by directly measuring [Na^+^]_i_ elevation, using the fluorescent probe ANG-2, only oxostephanine and liriodenine exhibited strong inhibitory effects against the hNa_V_1.3 channel, which was stably expressed in HEK293 cells. Moreover, in patch-clamp experiments, only bebeerine was able to efficiently block the Na^+^ currents elicited by the hNav1.2 and hNav1.6 channels. We also showed that by altering the conformation of Na_V_ channels, BTX allowed for the binding of oxostephanine and liriodenine.

The endocrine GH3b6 cell line has been extensively used as a cellular model to study Na_V_ channels in patch-clamp experiments, notably those expressed in the central nervous system [39]. As we have previously demonstrated, these cells mainly express the endogenous Na_V_1.3 channel subtype [35]. Thus, these cells represent a good compromise in the search for or in characterizing new ligands of the Na_V_1.3 channel, a Na_V_ channel involved in neuropathic pain [40]. To efficiently and rapidly screen our in-house IA library, we used VSP technology. Indeed, this strategy has been proven to be adapted for the detection and selection of ligands of Na_V_ channels from chemical libraries or natural fractions [41,42,43,44,45]. For this strategy, it is necessary to activate Na_V_ channels with selective activators, such as BTX or VTD. In our hands, fluorescent signals induced by BTX were stronger than those induced by VTD (data not shown), likely because BTX acts as a full activator [46]. However, despite an IC_50_ value in the nanomolar range for TTX, we were very surprised that BTX-induced VSP signals were inhibited at more than 50%, with almost half of our in-house library in the range of 12–20 µΜ. Those higher concentrations could lead to a non-specific action on the cell that would modify the VSP signal indirectly. We then decided to select only IA with an inhibition rate of more than 75%, which yielded 10 hits. Among these hits and according to the physico-chemical analysis, five IA were selected: oxostephanine, liriodenine, thalmiculine, bebeerine, and protopine.

The VSP assays indicated that these five IA inhibit the BTX-induced VSP signals in GH3b6 cells with different efficiencies. While the oxoaporphines, oxostephanine, and liriodenine totally suppressed the BTX-induced VSP signals, thalmiculine, bebeerine, and protopine hardly inhibited these signals to 30–65%. This could be explained by the fact that BTX activates the Na_V_ channel by binding to the pore with very high affinity, preventing any inactivation of the channels; this thereby promotes the identification of pore blockers, as it has already been discussed for VTD [47]. Thus, we first hypothesized that oxostephanine and liriodenine would behave as pore blockers, while bebeerine, thalmiculine, and protopine would exhibit a different mechanism of action. However, the use of VSP for the screening assay has been greatly debated, particularly in the search for selective Na_V_1.7 inhibitors [47,48]. Notably, this is because this technique requires the use of a neurotoxin activator, which can trap Na_V_ channels in a particular conformation. Thus, this approach needs complementary techniques, such as patch-clamp recording or the use of fluorescent Na^+^ probes, both of which directly measure Na_V_ channel activity. Here, we used ANG-2, which has been proven to be the most efficient in monitoring intracellular Na^+^ concentration [38].

Using this Na^+^ probe, we found that only oxostephanine and liriodenine were able to efficiently inhibit BTX-induced [Na^+^]_i_ elevations in hNa_V_1.3-HEK293 cells. Indeed, at 10 µM, these two IA almost abolished BTX-induced [Na^+^]_i_ elevations, suggesting the micromolar affinities of both IA for the hNa_V_1.3 channel in the presence of BTX. However, in our assay, thalmiculine, bebeerine, and protopine exhibited modest effects at 10 µM. Several hypotheses could be made: (1) thalmiculine, bebeerine, and protopine probably have weaker effects on the hNa_V_1.3 channel than oxostephanine and liriodenine, (2) thalmiculine, bebeerine, and protopine could block other Na_V_ channel subtypes, (3) the spatio-temporal resolution of our assay did not allow us to highlight their blocking effects on the hNa_V_1.3 channel, and (4) these three IA target other ion channels, such as the Cav and Kv channels, alleviating any change of [Na^+^]_i_ by the activation of NCX and other transporters. These data showed the advantages but also the limitations of using fluorescent probes in screening for Na_V_ channel ligands [38,47,49].

Our patch-clamp experiments showed that among the five IA selected by the VSP assays, only bebeerine behaved as a potent blocker of the hNa_V_1.2 and hNa_V_1.6 channels, with an IC_50_ value likely above 10 µM. Neither oxostephanine, liriodenine, thalmiculine, nor protopine had any inhibitory effects on the hNa_V_1.2, hNa_V_1.3, and hNa_V_1.6 channels at 10 µM, revealing that they are likely to be low-potency blockers of these Na_V_ channel subtypes. We could not use higher concentrations of these compounds because of their low solubility and the non-specific effects of the DMSO concentrations. These unexpected results could be explained by the bias induced by BTX as a Na_V_ activator in fluorescent-based assays, as it has been previously discussed for VTD [47]. Just as VTD, BTX is known to bind within the pore, thus trapping Na_V_ channels in an open state [8,13,50], allowing for the binding of low-affinity blockers [47]. This assumption was confirmed by the fact that oxostephanine and liriodenine decreased the BTX effects on both the activation and inactivation properties of the hNa_V_1.3 channel, while both IA had no effects on the inactivation properties of this Na_V_ channel in the absence of BTX. Thus, we assumed that both IA bind to Na_V_ channels in the presence of BTX, which traps Na_V_ channels in an open state. Altogether, our data highlighted that fluorescent-based screening assays on BTX-activated Na_V_ channels could compromise the finding of potent and selective blockers of these ion channels, and that the patch-clamp technique remains unavoidable for the accurate characterization of Na_V_ channel ligands.

Oxostephanine and liriodenine are two members of the oxoaporphine subgroup, and they only differ by a methoxy group on the benzyl part of the skeleton of this IA subgroup. Previous studies have demonstrated the ability of liriodenine to suppress Na^+^ currents recorded in isolated cardiomyocytes with an IC_50_ value in the micromolar range [27,51], which is close to what we observed. Since cardiomyocytes mainly express Na_V_1.5 [2], it is possible that liriodenine preferentially blocks this Na_V_ channel subtype. Several studies have shown that oxostephanine and liriodenine share some biological activities, such as antimicrobial and anti-cancer properties [52,53]. Liriodenine has been proven to target ion channels such as the K_V_, Ca_V,_ and Na_V_ channels [27], but it also targets receptors such as the GABAA, nicotinic, and muscarinic receptors [54,55,56]. The high structural similarity between oxostephanine and liriodenine certainly supports the fact that they might share the same pharmacological targets, such as the Nav channels.

Finally, for the first time, we determined that bebeerine exhibits inhibitory effects on Na_V_ channels, with a higher potency on the hNa_V_1.6 channel than on the hNa_V_1.2 and hNa_V_1.3 channels. Bebeerine is the enantiomer of curine, which has been extensively studied and previously described as an L-Type Ca_V_ channel blocker [57]. Thus, it is conceivable that bebeerine might also inhibit L-Type Ca_V_ channels. Interestingly, no BBIQ has been reported as a Na_V_ channel blocker so far, and since this IA subgroup is particularly large, further screening by an automated patch-clamp could be of interest. In addition, further studies are required to provide more information on the selectivity and also the blocking mechanism of bebeerine towards all Na_V_ channel subtypes.

## 4. Materials and Methods

### 4.1. Chemicals

A voltage sensor probe (VSP) set (CC2-DMPE and DisBAC_2_(3)), VABSC-1 (Voltage Assay Background Suppression Compound), and Pluronic^®^-F127 were purchased from Invitrogen (Carlsbad, CA, USA). BTX and TTX were purchased from Latoxan (Valence, France). Asante NaTRIUM Green-2 (ANG-2) was purchased from Interchim (Montluçon, France). All other reagents (Ponceau 4R) and solvents were obtained from Sigma-Aldrich (Saint-Louis, MO, USA) or Fisher Scientific (Waltham, MA, USA).

### 4.2. In-House Library of Vegetal Alkaloids

More than 300 alkaloids from plants have been collected in a library of the SONAS laboratory (EA 921, University of Angers) in the last four decades. These compounds were extracted and purified from at least 14 plant families (*Annonaceae*, *Berberidaceae*, *Fumariaceae*, *Lauraceae*, *Menispermaceae, etc*.), which were collected from all over the world (Bolivia, Colombia, France, Indonesia, Jordan, etc.) (Appendix A). They were stored dried at room temperature. In a blind test, 62 IA associated with 10 different structural subtypes (Appendix A) were screened for their ability to inhibit BTX-evoked depolarization in GH3b6 cells. Each IA or fraction was solubilized in DMSO at 1–5 mg/mL, stored, and sheltered from light at 4 °C.

### 4.3. Cell Culture

GH3b6 cells are a subclone of the GH3 pituitary cell line [58] and were a generous gift from Dr. Françoise Macari (IGF, Montpellier, France). GH3b6 cells were cultivated from passage 14 until passage 32 at 37 °C/5% CO_2_ in Dubelcco’s Modified Eagle Medium (DMEM)/F12 (Lonza, Basel, Switzerland), supplemented with 10% FBS (Lonza, Basel, Switzerland), 1 mM L-glutamine, and 1 mM penicillin and streptomycin solution. For the fluorescence assay, the GH3b6 cells were seeded in black-walled, clear-bottom, 96-well, tissue culture-treated, and sterile imaging plates (Corning Hazebrouck, France), at a density of 10,000 cells per well, and incubated for 24 h before starting the experiment. The human Na_v_1.3 (hNa_v_1.3) channel-expressing cell line (HEK293) used in this work was a generous gift from Massimo Mantegazza (IPMC, Nice, France). HEK293-hNa_V_1.3 cells were cultivated from passage 20 until passage 35 at 37 °C/5% CO_2_ in a DMEM high glucose medium (4.5 g/l), supplemented with 10% FBS, pyruvate, L-glutamine, and penicillin and streptomycin solution, each one at 1 mM, completed with 800 µg/mL geneticin G418. For the fluorescence assays, the cells were plated at a density of 50,000 cells per well in black-walled, clear-bottom, 96-well plates and then cultured at 37 °C and 5% CO_2_ for 24 h before the experiments. HEK293 stably expressing hNa_V_1.2 and hNa_V_1.6 were cultivated in the same conditions mentioned above for the HEK293-hNa_V_1.3 cells. For the automated patch-clamp experiments, cells were passaged from 24 to 48 h prior to the experiment and replated in the appropriate quantity to reach 50–70% confluence. For electrophysiological recordings, cells were detached with trypsin-EDTA and diluted in a culture medium. Then, the cells were centrifuged at 800 rpm for 4 min and resuspended (~300,000 cells/mL) in a patch-clamp extracellular solution.

### 4.4. Fluorescent Assays

Voltage sensor probe (VSP) assays were carried out as described previously [59]. VSP were freshly prepared in an Na-free buffer (VSP Buffer) containing (in mM): 160 TEA-Cl, 0.1 CaCl_2_, 1 MgCl_2_, 11 Glucose, and 10 HEPES-K (pH 7.4). Cells were washed with the VSP Buffer and first incubated with CC2-DMPE (5 µM) and Pluronic^®^-F127 acid (0.01%) dissolved in the VSP Buffer for 70 min at RT. After washing with the VSP buffer, the cells were loaded with DisBAC_2_(3) (10 µM) and VABSC-1 (250 µM) dissolved and diluted in the VSP Buffer for 45 min at RT. In control experiments, BTX (1–10 µM) with and without TTX (1–10 µM) was added to the VSP buffer containing DisBAC_2_(3) and VABSC-1. The plates were illuminated at λ = 405 nm, and the resulting fluorescence emissions were monitored at λ = 460 and λ = 580 nm. The plate reader measured the fluorescence emission every 5 s up to 145 s. After a 30 s baseline, cell depolarization was induced by the injection of a depolarizing solution (in mM: 160 NaCl, 4.5 KCl, 2 CaCl_2_, 1 MgCl_2_, 11 glucose, 10 HEPES-K, (pH 7.4)), raising the final Na^+^ concentration to 80 mM. The osmolarity of the solution remained the same at around 300 ± 5 mOsmol before and after the addition of the depolarizing solution. For the screening assays, IA were co-incubated at 10–50 µg/mL with 1 µM of BTX in triplicate. In positive control experiments, BTX (1 µM) with TTX (10 µM) and DMSO (1%) were added to the VSP buffer containing DisBAC_2_(3) and VABSC-1. To establish the concentration–response relationships, increasing concentrations of compounds were co-incubated with BTX (1 µM). These experiments were performed in three wells and repeated twice. Intracellular Na^+^ fluorescent assays were performed using the cell-permeant acetoxymethyl esters (AM) form of the Asante NaTRIUM Green-2 (ANG-2) probe, a cytosolic Na^+^ indicator. The HEK293-hNa_v_1.3 cells were incubated for 1 h in the dark at 37 °C in the ANG-2 probe (5 µM) prepared with Hank’s buffer saline solution (HBSS), which contained (in mM): 2 CaCl_2_, 1 MgCl_2_, 5 KCl, 150 NaCl, 10 D-glucose, 10 HEPES (pH 7.4), and 0.02% pluronic acid. Afterwards, the quencher solution Ponceau 4R (P4R, 1 mM) was added, and the fluorescence signal was monitored at the following wavelengths: excitation 517 nm and emission 540 nm. Thirty seconds after baseline recording, BTX (1 µM) was automatically injected with or without TTX (1 µM) or IA (10 µM), and the fluorescence emission was read for 1000 s. All experiments were performed in triplicate wells and repeated thrice. Fluorescence measurement was carried out using a FlexStation^®^ 3 Benchtop Multi-Mode Microplate Reader (Molecular Devices, Sunnyvale, CA, USA).

### 4.5. Quantitative Real Time PCR

Total RNA from the GH3b6 cells was extracted using the RNeasy^®^ micro kit (Qiagen, Courtaboeuf, France). For retrotranscription, 1 µg of total RNA was used, and the reactions were performed using random hexamer primers and the QuantiTect^®^ Reverse Transcription kit (Qiagen). Real-time PCR assays were carried out on a LightCycler^®^ 480 Instrument II (Roche, Meylan, France) using the Sybr^®^ Select Master Mix (Applied Biosystems) with 2.5, 5, and 10 ng of cDNA in duplicate. Gene-specific primers for real-time PCR (Table 4) were designed using the Primer3 Software. Differences in transcript levels were determined using the cycle threshold method, as described by the manufacturer. Specificity of amplification was confirmed by melting curve analysis. Relative quantification of gene expression was normalized to the mean of expression of two validated House-keeping genes: *gapdh* (glyceraldehyde-3-phosphate dehydrogenase) and *Gusb* (beta-glucuronidase), according to the formula E = 2^−(Ct(Target)−Ctmean(Reference))^, where C_t_ is the threshold cycle. Amplicon sizes (70–106 bp) and AT content (47–55%) were chosen to allow for a comparison of the relative expression values obtained for each gene [60].

### 4.6. HPLC Analysis

Chromatographic analyses were performed on a 2695 separation module coupled to a 2695 PDA detector (Waters, Saint Quentin en Yvelines, France), assisted by the Empower software (Waters). The IA were dissolved in MeOH at 1 mg/mL and centrifuged for 6 min at 13,000 rpm. The injection volume was 5 µL. Samples were injected into a Lichrospher 100 RP18 column (125 mm × 4 mm, 5 µm, Merck) and were separated by a gradient elution with buffer A (50 mM KH_2_PO_4_, 2 mM heptane sulfonic acid, 0.1% diethylamine, pH = 3) and the MeCN (B) solvent system (T_0_ 5% B; T_25′_ 100% B and T_30′_ 100% B) [61]. The flow rate was 1 mL/min, with UV detection at 210 and 280 nm, respectively.

### 4.7. Automated Patch-Clamp Experiments

Whole-cell recordings were used to investigate the effects of oxostephanine, liriodenine, thalmiculine, bebeerine, and protopine on HEK293 cells stably expressing the hNav1.2, hNav1.3, and hNav1.6 channels. Automated patch-clamp recordings were performed using the SyncroPatch 384PE from Nanion (München, Germany). Chips with a single hole (n = 384, series resistance 3.93 ± 1.19 MΩ) were used for HEK293 cell recordings. Pulse generation and data collection were performed with the PatchControl384 v1.9.7 software (Nanion) and the Biomek interface (Beckman Coulter). Whole-cell recordings were conducted according to the recommended procedures of Nanion. The cells were stored in a cell hotel reservoir at 10 °C, with a shaking speed of 60 RPM. After initiating the experiment, cell catch, sealing, whole-cell formation, liquid application, recording, and data acquisition were all performed sequentially and automatically. The intracellular solution contained (in mM): 10 CsCl, 110 CsF, 10 NaCl, 10 EGTA, and 10 HEPES (pH 7.2, osmolarity 280 mOsm), and the extracellular solution contained (in mM): 60 NMDG, 80 NaCl, 4 KCl, 2 CaCl_2_, 1 MgCl_2_, 5 Glucose, and 10 HEPES (pH 7.4 (NaOH), osmolarity 280 ± 3 mOsm). Whole-cell experiments were performed at a holding potential of −100 mV at room temperature (18–22 °C). Currents were sampled at 20 kHz. For pharmacological assays, alkaloid solutions were prepared at 30 µM in the extracellular solution supplemented with 0.3% bovine serum albumin and distributed in 384-well compound plates according to a pre-established plate plan. This distribution within compound plates could not be randomized since it would then excessively complexify the methods of analyses. The working compound solution was diluted 3 times in the patch-clamp recording well by adding from 30 to 60 µL of the external solution to reach the final reported concentration and the test volume of 90 µL. The effects of IA (10 µM) or of BTX (1 µM) were measured at the end of a 10 min application time. Activation curves were built by depolarizations lasting for 2000 ms, from −130 mV to 40 mV, using 5 mV increment steps. Pulses were applied every 5 s. The use dependency of oxostephanine and the liriodenine effects on hNa_V_1.3 with or without the presence of 1 µM BTX were investigated by depolarizing the cells to 10 mV at 1 Hz for 300 s.

### 4.8. Structural Analysis

^1^H NMR spectra were conducted in the appropriate deuterated solvent (CHCl_3_-d or CH_3_OH-d_4_) on a Bruker Avance DRX-500 spectrometer (Bruker France, Wissembourg, France) operating at 500 MHz (^1^H). Fast Atom Bombardment (FAB) mass spectra were obtained on a B/E JMS 700 MStation spectrometer (Jeol Europe, Croissy-sur-Seine, France), with 3-nitrobenzyl alcohol as the matrix and argon as the collision gas. Optical rotations (${\left[\alpha \right]}_{D}^{T{}^{\circ}C}$) were measured at 20 °C on a Polartronic I polarimeter (Schmidt-Haensch, Berlin, Germany).

### 4.9. Data Bank Search

Structural analogs of oxostephanine and liriodenine were searched for using the search tool from ChemIDplus (https://chem.nlm.nih.gov/chemidplus/ProxyServlet, June 2021). Analogs sharing less than 94% were ignored.

### 4.10. Data Analysis

Fluorescence data analysis was performed using the SoftMax Pro 5.4.1 software (Molecular Devices, Sunnyvale, CA, USA).

With data from VSP assays, calculations made for histograms were performed as follows. For t = 1 s up to 30 s, for each condition, the VSP signal ratio of the GH3b6 cells was measured after 30 min of incubation with 1 µM BTX and an individual alkaloid. The first 30 s showed signal at equilibrium before the addition of the depolarizing solution, resulting in a final Na^+^ concentration of 80 mM. Each curve point is the CC2DMPE signal over the DisBAC(2,3) ratio, as a ratio relative to the response before the addition of the depolarizing solution (calculated by averaging the first 4 time points, T1-4, from each replicate). The baseline of this ratio was shifted to 0 by subtracting the average of the 4 first time points before adding the depolarizing solution. Each value was then expressed as a percentage of the positive control final ratio obtained in each plate/experiment with 1 µM BTX. This allowed us to compare data from different microplates. The equation used was the following:$$\frac{\left(\frac{\left(CC2DMPE/DisBAC2,3\right)}{Average_{T1-4}\left(CC2DMPE/DisBAC2,3\right)}-1\right)\times 100}{\left(\frac{AverageControlBTX_{T27-30}\left(CC2DMPE/DisBAC2,3\right)}{AverageControleBTX_{T1-4}\left(CC2DMPE/DisBAC2,3\right)}-1\right)}$$

These values were plotted over time (Figure 4a). The histograms (Figure 4b) represent the inhibitory effect of each IA, normalized as follows: 100% (VSP signal induced by BTX at 1 µM) average of the last 4 values of each curve.

The kinetic traces of the ANG-2 fluorescence were plotted as an emission ratio: λexcitation 517 nm/λexcitation 540 nm. The AUC of the kinetic traces were used to determine the response after BTX injection, alone (1 µM) or with each IA (10 µM).

All graphs and statistical analyses were completed using the GraphPad Prism 7.02 (La Jolla, CA, USA). Data are presented as mean ± SEM, calculated from 3 replicates and representative of 3 independent experiments. Non-linear analysis was used to fit the concentration–response data with the Langmuir–Hill equation with a variable slope. Multiple groups were compared by using a one-way analysis of variance (ANOVA), followed by a Bonferroni post hoc test, when appropriate. Differences between independent groups were assessed by using parametric or non-parametric *t*-tests (Mann and Whitney) when appropriate. Differences with *p* < 0.05 were considered significant (* for *p* < 0.05, ** for *p* < 0.01, *** for *p* < 0.001, **** for *p* < 0.0001).

## 5. Conclusions

In conclusion, the VSP screening of the in-house alkaloid library allowed for the identification of five IA—oxostephanine, liriodenine, thalmiculine, bebeerine, and protopine—as inhibitors of BTX-activated Na_V_ channels. However, only bebeerine was validated as a potent blocker of Na_V_ channels by automated patch-clamp experiments, with the following rank order: hNa_V_1.6 > hNa_V_1.3 > hNa_V_1.2. In fact, oxostephanine and liriodenine were able to decrease the BTX-induced alteration of both the activation and inactivation properties of the hNa_V_1.3 channel. Our data highlight that the use of the Na_V_ channel activator, which binds within the pore, could lead to the selection of low-potency Na_V_ channel inhibitors, and that electrophysiology is unavoidable for their characterization. Finally, bebeerine is the first BBIQ exhibiting potent inhibitory effects on the hNa_V_1.3 and hNa_V_1.6 channels.

## Figures and Tables

**Figure 1 molecules-27-04133-f001:**
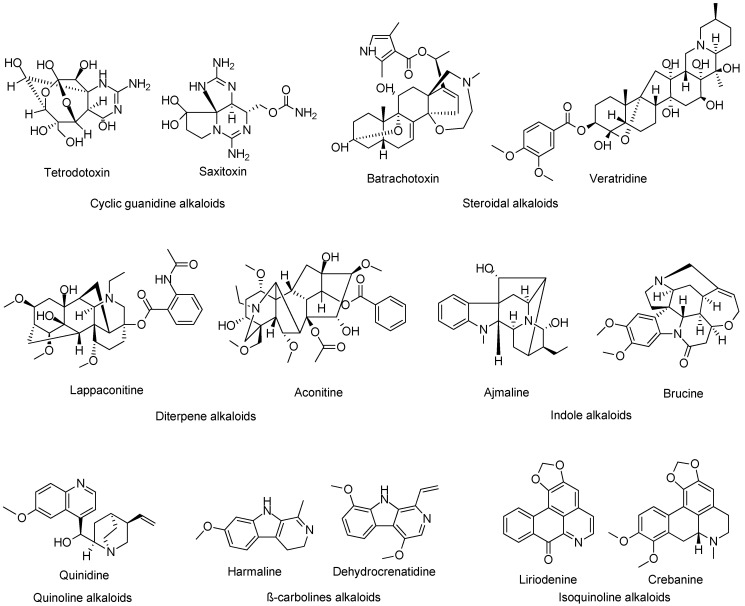
Structural diversity of natural alkaloids active on Na_V_ channels. Tetrodotoxin and saxitoxin are two marine guanidine alkaloids. Veratridine, aconitine, and batrachotoxin are activators of the Na_V_ channels. Lappaconitine, quinidine, and ajmaline are two inhibitors used as clinical drugs. Liriodenine and crebanine, two IA, block the Na_V_ channels. Lappaconitine, which is structurally related to aconitine [15], exhibits potent analgesic properties [16] and has recently been proposed for the treatment of chronic pain [17]. Ajmaline, isolated from *Rauwolfia serpentina,* inhibits the Na_V_1.5 channels [18,19,20,21]. Ajmaline is used in the diagnosis and treatment of cardiac arrhythmia [22]. Brucine, extracted from *Nux vomica*, exhibits antinociceptive properties mediated by Na_V_ channel inhibition [23]. Harmaline from *Peganum harmala* [24] and dehydrocrenatidine from *Picrasma quassioides* are used to treat pain [25].

**Figure 2 molecules-27-04133-f002:**
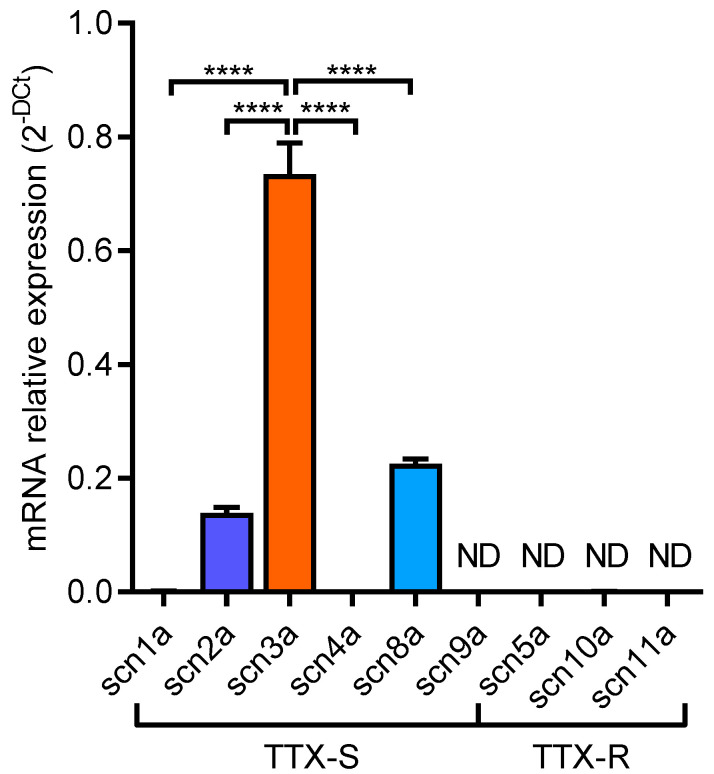
GH3b6 cells express TTX-S Nav channels. Nav channel gene expression profile of GH3b6 cells. The histogram illustrates the expression repertoire of Nav channel genes in GH3b6 cells as established by RT-qPCR. Data are the mean of mRNA relative expression ± SEM (*n* = 6). ND: Not Detected. Significance tests between groups of data were performed using one-way ANOVA followed by Bonferroni’s multiple comparison test (**** *p* < 0.0001).

**Figure 3 molecules-27-04133-f003:**
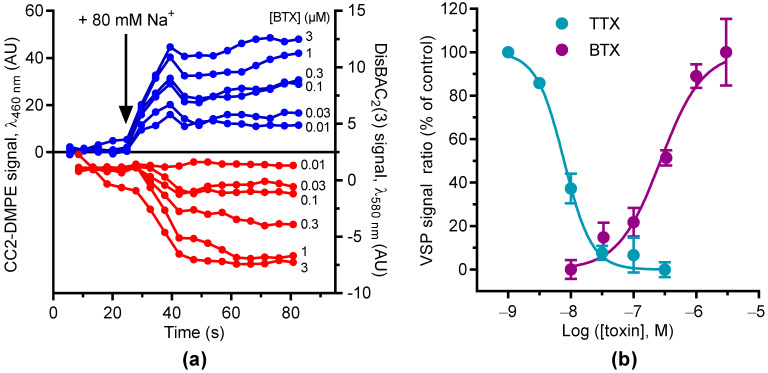
Monitoring the BTX activation of Na_V_ channels using VSP in GH3b6 cells. (**a**) Representative kinetic traces illustrating the fluorescence emission of CC2-DMPE (λ = 460 nm, blue line) and DisBAC2(3) (λ = 580 nm, red line) in the presence of increasing concentrations of BTX (from 0.01 to 3 µM) before and after raising the Na^+^ concentration to 80 mM. Data are the mean of three wells. (**b**) Concentration–response curves of a BTX-induced VSP signal. BTX induced an increase in VSP signal ratio in a concentration-dependent manner. TTX decreased the VSP signal ratio induced by BTX at 1 µM in a concentration-dependent manner. Normalized data were calculated with four end points of the fluorescence emission ratio (Signal ratio CC2-DMPE/DisBAC2(3)) kinetics. Data were best fitted with the Hill–Langmuir equation with a variable slope. For BTX, EC_50_ = 0.27 ± 0.06 µM, Hill slope = 1.21 ± 0.26. For TTX, IC_50_ = 8.58 ± 1.67 nM, Hill slope = 1.53 ± 41. Data are mean ± SEM (*n* = 3).

**Figure 4 molecules-27-04133-f004:**
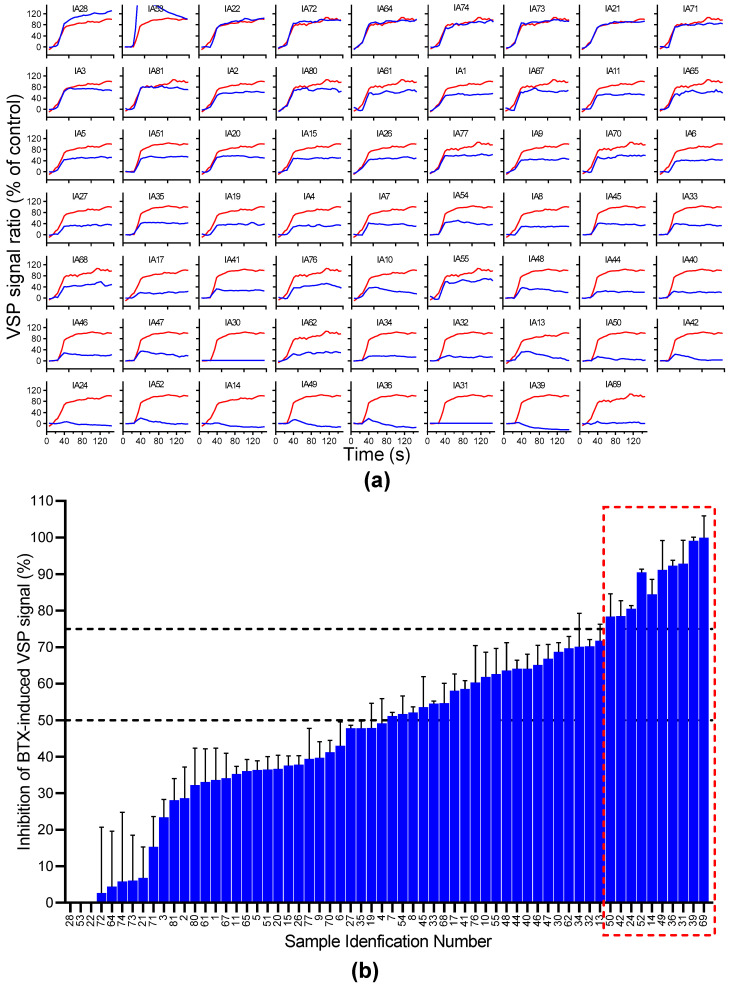
Screening of 62 samples from an in-house library of plant alkaloids for inhibitors of Na_V_ channels. (**a**) Average kinetic ratios of VSP signal obtained in the presence of BTX alone (red line) and BTX co-incubated with IA (blue line), before and after injection of the depolarizing solution (at 30 s). Red and blue traces correspond to distinct experiments. x axis: fluorescence ratio (a.u.), y axis: time (s). (**b**) The histogram bars correspond to the inhibition of the BTX-induced VSP signal (in %). Ten IA induced at least 75% of the inhibition of BTX-induced depolarization (red dashed rectangle). Data are mean ± SEM (*n* = 3).

**Figure 5 molecules-27-04133-f005:**
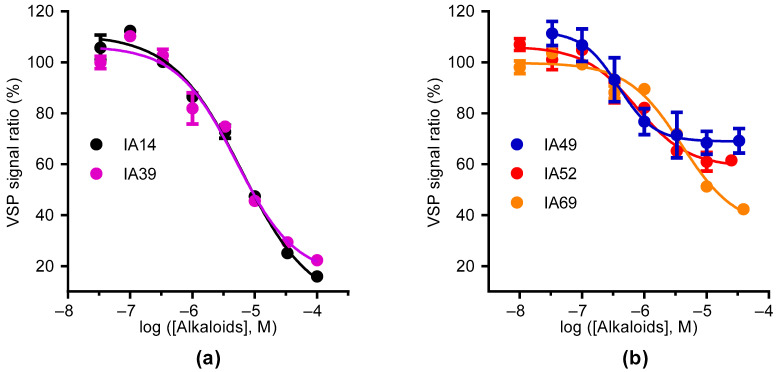
Concentration–inhibition curves of oxostephanine, liriodenine, thalmiculine, bebeerine, and protopine for the BTX-induced VSP FRET signal in GH3b6 cells. (**a**) Concentration–inhibition responses of the BTX-induced VSP signal to increasing concentrations of oxostephanine (IA14) and liriodenine (IA39). (**b**) Concentration–inhibition responses of the BTX-induced VSP signal to increasing concentrations of thalmiculine (IA49), bebeerine (IA52), and protopine (IA69). The data were best fitted by non-linear regression to the Langmuir–Hill equation with a variable slope and are shown in Table 2. Data are mean (± SEM) and representative of two independent experiments (*n* = 6).

**Figure 6 molecules-27-04133-f006:**
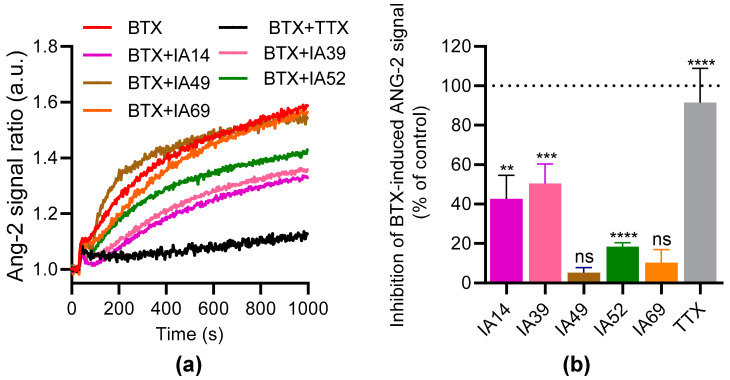
Effects of oxostephanine, liriodenine, thalmiculine, bebeerine, and protopine on the ANG-2 signal induced by BTX in HEK293-hNa_V_1.3 cells. (**a**,**b**) Representative kinetic traces illustrating the fluorescence emission of ANG-2 probe induced by the injection of BTX (3 µM) without or with oxostephanine (IA14), liriodenine (IA39), bebeerine (IA52), thalmiculine (IA49), and protopine (IA69) at 10 µM each. TTX (1 µM) was used as control. (**c**,**d**) The histogram bars represent the inhibition rate (in %) of each IA at 10 µM on the ANG-2 fluorescence signal induced by BTX. Normalized data were calculated as described for the VSP signal. Data are mean ± SEM of 3–5 independent experiments. Significance was evaluated by two-tailed unpaired *t*-tests between each group, with zero as a hypothetical value. **, *p* < 0.01; ***, *p* < 0.001; ****, *p* < 0.0001; ns: not significant.

**Figure 7 molecules-27-04133-f007:**
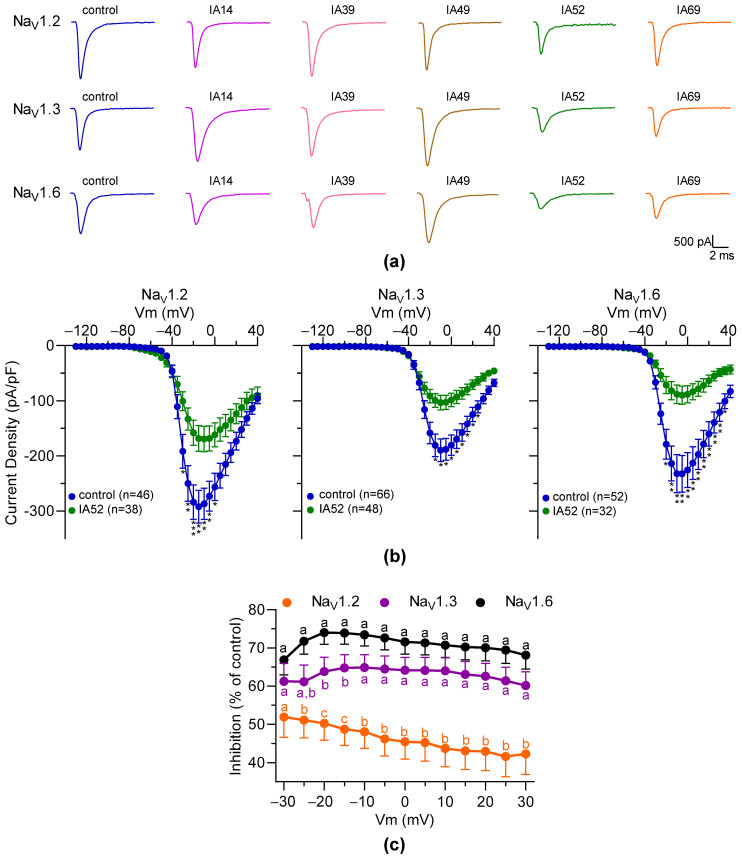
Effects of oxostephanine, liriodenine, thalmiculine, bebeerine, and protopine on Na^+^ currents elicited by the hNa_V_1.2, hNa_V_1.3, and hNa_V_1.6 channels. Oxostephanine (IA14), liriodenine (IA39), thalmiculine (IA49), bebeerine (IA52), and protopine (IA69) were assayed at 10 µM each on the hNa_V_1.2, hNa_V_1.3, and hNa_V_1.6 channels stably expressed in HEK293T cells. Control experiments were performed with an extracellular solution containing 0.1% of DMSO corresponding to the vehicle. (**a**) Examples of Na^+^ current traces elicited at +20 mV in the presence and in the absence of IA14, IA39, IA49, IA52, and IA69. (**b**) I-V relationship curves obtained by plotting the mean of peak current density to Vm. Data are shown as mean ± SEM. Multiple unpaired *t*-test corrected with the Holm–Sidak method was used to compare groups. Only significant results are shown. *, *p* < 0.05; **, *p* < 0.001; ***, *p* < 0.0001. (**c**) Comparison of the inhibitory effects of IA52 on the hNa_V_1.2, hNa_V_1.3, and hNa_V_1.6 channels. The inhibition rate induced by IA52 on the Na^+^ currents was normalized (Vm = −30 to 30 mV). Data were plotted as mean ± SEM and analyzed for statistical significance using a two-way ANOVA test with Holm–Sidak’s multiple comparison test (*p* < 0.0001). Different letters (a, b, and c) above (hNa_V_1.2 and hNa_V_1.6) and below (hNa_V_1.3) indicate significant differences between groups. *p*-values are shown in Appendix A.

**Figure 8 molecules-27-04133-f008:**
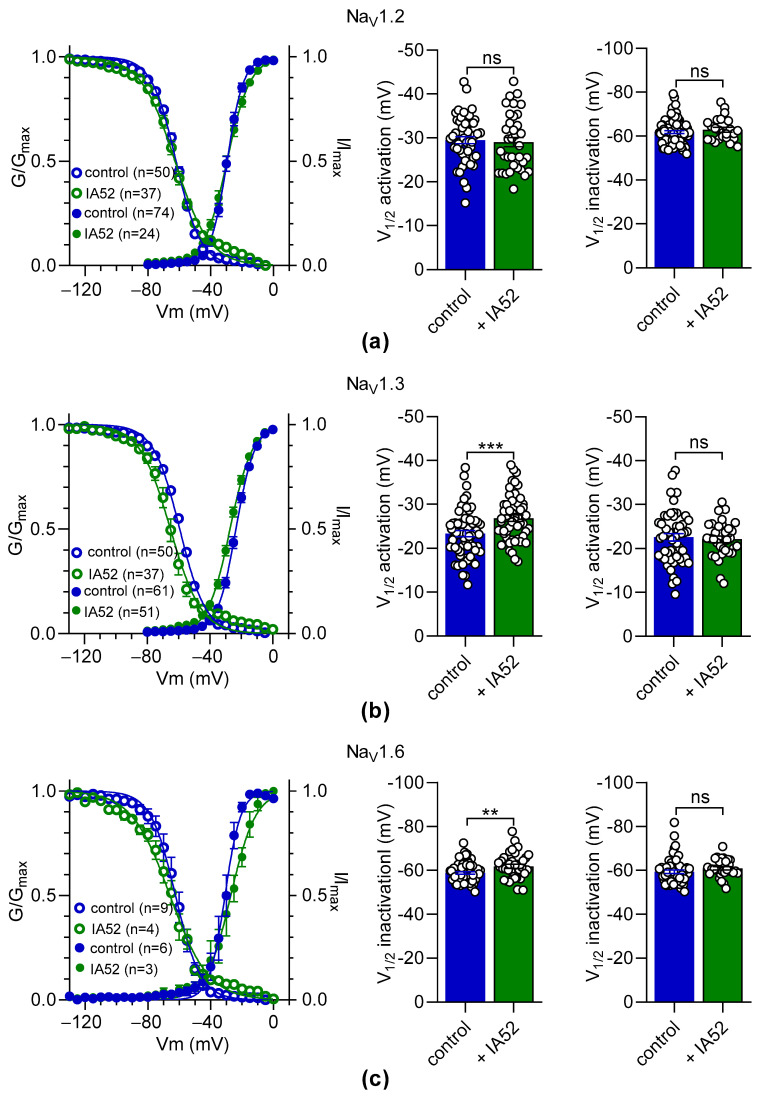
Effects of bebeerine on the activation and inactivation properties of Na^+^ currents elicited by the hNa_V_1.2, hNa_V_1.3, and hNa_V_1.6 channels. The effects of bebeerine (IA52, 10 µM) on the voltage-dependent activation and inactivation of the hNa_V_1.2 (**a**), hNa_V_1.3 (**b**), and hNa_V_1.6 (**c**) channels were evaluated. Right panels (**a**–**c**): Activation and inactivation curves show data that were fit with the Boltzmann equation. Middle and left panels: The scatter plots with bars show the V_1/2_ of the activation and inactivation of the hNa_V_1.2 (**a**), hNa_V_1.3 (**b**), and hNa_V_1.6 (**c**) channels. Control experiments were performed with extracellular solution containing 0.1% of DMSO corresponding to the vehicle. Data are shown as mean ± SEM. An unpaired *t*-test was used to analyze the significance between groups. ns: not significant; **, *p* < 0.01; ***, *p* < 0.001. Fitted parameters are shown in Table 3.

**Figure 9 molecules-27-04133-f009:**
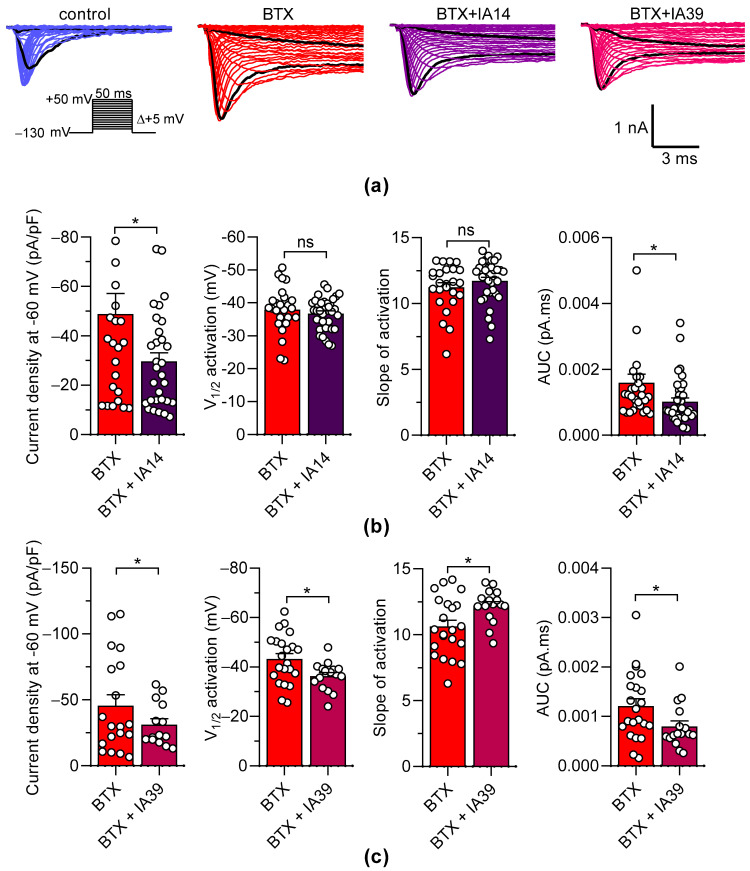
Effects of oxostephanine and liriodenine on Na^+^ currents elicited by hNa_V_1.3 in the presence of batrachotoxin. Oxostephanine (IA14) and liriodenine (IA39) were assayed at 10 µM each on Na^+^ currents elicited by the hNa_V_1.3 channel in the presence of batrachotoxin (BTX, 1 µM). (**a**) Representative examples of Na^+^ currents elicited by 50 ms depolarization steps (protocol inset). To illustrate the strong effect of 1 µM BTX, traces at −60 mV and 20 mV are shown in black. (**b**,**c**) The scatter plots with bars illustrate: the current densities at −60 mV, the V_1/2_ of activation, the slope of activation curves, and the area under the curve (AUC in pA.ms) in the absence or in the presence of IA14 (**b**) and IA39 (**c**). Data are shown as mean ± SEM. Parametric or non-parametric statistical analyses were performed to determine the significance between groups. *, *p* < 0.05; ns: not significant.

**Table 1 molecules-27-04133-t001:** Chemical analysis of IA selected for their ability to inhibit a BTX-induced VSP signal.

Sample	Inhibitory Effects (%)	Alkaloid Name	Plant Name	210 nm * (%)	280 nm * (%)	^1^HNMR, $\left(\mathrm{MS},\;{\left[\alpha \right]}_{D}^{20{}^{\circ}C}\right)$	Acceptance
IA50	78.3	tetrandine	*Pachygone dasycarpa (Menispermaceae)*	76	67	ND	No
IA42	78.5	gyrocarpusine	*Gyrocarpus americanus* *(Hernandiaceae)*	68	52	ND	No
IA24	80.5	oxostephanine	*Stephania venosa* *(Menispermaceae)*	77	98	Confirmed	Yes
IA52	81.9	bebeerine	*Curarea candicans (Menispermaceae)*	83	100	Confirmed	Yes
IA14	84.4	oxostephanine	*Stephania venosa* *(Menispermaceae)*	91	99	Confirmed	Yes
IA49	91.1	thalmiculine	*Thalictrum cultratum (Ranunculaceae)*	93	85	Confirmed	Yes
IA36	92.2	sukhodianine	*Stephania venosa* *(Menispermaceae)*	65	81	ND	No
IA31	92.8	tiliacorinine	*Tiliacora racemosa* *(Menispermaceae)*	73	81	ND	No
IA39	99.1	liriodenine	*Stephania venosa* *(Menispermaceae)*	77	99	Confirmed	Yes
IA69	99.9	protopine	*Corydalis majori* *(Fumariaceae)*	85	75	confirmed	Yes

This table shows the list of IA which inhibited a BTX-evoked VSP signal at more than 65% in GH3b6 cells. * Each compound was analyzed by HPLC monitored at two different wavelengths (210 and 280 nm) in order to evaluate their purity in %; below 75%, the compound was rejected. Then, the NMR-^1^H (sometimes MS and ${\left[\alpha \right]}_{D}^{20{}^{\circ}C}$) was performed to confirm the IA structure. ND: not determined.

**Table 2 molecules-27-04133-t002:** Curve fitting parameters of the concentration–inhibition relationships of oxostephanine, liriodenine, thalmiculine, bebeerine, and protopine.

Sample	Alkaloid Name	Alkaloid Type	IC_50_ (µM)	Hill Slope	Maximum of Inhibition
IA14	oxostephanine	oxoaporphine	6.11 ± 0.09	0.78 ± 0.12	95.78 ± 6.99
IA39	liriodenine	oxoaporphine	5.00 ± 0.10	0.91 ± 0.19	83.78 ± 6.59
IA49	thalmiculine	BBIQ	0.38 ± 0.19	1.50 ± 0.93	31.07 ± 4.04
IA52	bebeerine	BBIQ	0.74 ± 0.11	1.04 ± 0.29	41.04 ± 3.44
IA69	protopine	protopine	3.97 ± 0.10	0.99 ± 0.19	64.67 ± 5.50

This table shows the IC_50_, Hill coefficient, and the maximum of inhibition values determined from non-linear regression analysis of concentration–inhibition curves of BTX-evoked VSP signal by oxostephanine, liriodenine, thalmiculine, bebeerine, and protopine of Figure 5. Data are mean ± SEM (*n* = 6).

**Table 3 molecules-27-04133-t003:** Fitting parameters of the voltage-dependent activation and inactivation of Na^+^ currents elicited by the hNa_V_1.2, hNa_V_1.3, and hNa_V_1.6 channels in the absence and in the presence of bebeerine.

		Activation		Inactivation	
Na_v_ Channel Subtype	Condition	V_1/2_ (±SEM, mV)	Slope (±SEM)	V_1/2_ (±SEM, mV)	Slope (±SEM)
hNa_V_1.2	control	−29.5 ± 0.8 (n = 48)	3.9 ± 0.2	−61.8 ± 0.7 (n = 66)	−6.4 ± 0.1
	+bebeerine	−29.1 ± 1.1 (n = 36)	5.8 ± 0.3 ^1^	−62.3 ± 1.1 (n = 21)	−8.5 ± 0.3 ^4^
hNa_V_1.3	control	−23.3 ± 0.7 (n = 61)	4.9 ± 0.2	−58.2 ± 0.6 (n = 54)	−7.1 ± 0.2
	+bebeerine	−26.9 ± 0.7 (n = 51)	6.4 ± 0.3 ^2^	−60.2 ± 0.9 (n = 26)	−8.9 ± 0.3 ^5^
hNa_V_1.6	control	−22.5 ± 0.8 (n = 52)	5.1 ± 0.2	−59.0 ± 0.8 (n = 50)	−7.7 ± 0.3
	+bebeerine	−22.1 ± 0.7 (n = 32)	7.0 ± 0.3 ^3^	−60.2 ± 0.7 (n = 24)	−9.0 ± 0.3 ^6^

This table shows the fitting parameters determined from the non-linear regression analysis of the voltage-dependent activation and inactivation of Na^+^ currents in the absence or presence of 10 µM bebeerine (IA52, Figure 8). V_1/2_, half potential (mV) for voltage-dependent activation or inactivation. Data are means ± SEM. Statistical significance between the control and test groups was evaluated with an unpaired *t*-test. ^1,2,3,4,5^, *p* < 0.0001; ^6^, *p* < 0.001.

**Table 4 molecules-27-04133-t004:** Primers pairs used to amplify Nav channel transcripts by RTq-PCR.

Gene Name	GenBank Accession Number	Protein Name	Forward Primer (5’–3’)	Reverse Primer (5’–3’)	Amplicon Size (bp)
*scn1a*	NM_030875.1	Na_V_1.1	gttccgacatcgccagtt	catctcagtttcagtagttgttcca	92
*scn2a*	NM_012647.1	Na_V_1.2	tggtgtccctggttggag	ccttatttctgtctcagtagttgtgc	88
*scn3a*	NM_013119.1	Na_V_1.3	gcaccgtccattctaaccat	tttagcttcttgcataagaattgc	92
*scn4a*	NM_013178.1	Na_V_1.4	ggcactgtctcgatttgagg	ttcatgatggaggggatagc	71
*scn5a*	NM_013125.2	Na_V_1.5	tgccaccaatgccttgta	catgatgagcatgctaaagagc	96
*scn8a*	NM_019266.2	Na_V_1.6	ggaagttttccatcatgaatcag	gctgttatgtcgggagagga	70
*scn9a*	NM_133289	Na_V_1.7	cagcagatgttagaccgactca	actcgtgaactcagcagcag	78
*scn10a*	NM_017247.1	Na_V_1.8	agaggaccccaagggaca	tggtggttttcacacttttgg	59
*scn11a*	NM_019265.2	Na_V_1.9	cagaggacgatgcctctaaaa	ttctgggacagtcgtttggt	60
*gusb*	NM_017015.2	Gus	ctctggtggccttacctgat	cagactcaggtgttgtcatcg	78
*gapdh*	NM_017008.3	Gapdh	tgggaagctggtcatcaac	gcatcaccccatttgatgtt	77

Primer pairs used to amplify Na_V_ channel transcripts by RTq-PCR. GenBank accession number, sequence, and amplicon size are listed. The primer sequences for the housekeeping gene *gusb* and *gapdh* are also indicated.

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
