# Peer review of "Screening an In-House Isoquinoline Alkaloids Library for New Blockers of Voltage-Gated Na+ Channels Using Voltage Sensor Fluorescent Probes: Hits and Biases"

_molecules, 2022, doi:10.3390/molecules27134133_

Round 1

Reviewer 1 Report

The present study was to explore whether oxostephanine and liriodenine have possible effects on voltage-gated sodium channels in GH3b6 cells. There are several queries regarding the present investigations which are shown below.

  1. The fluorescent assays with voltage sensor probe (VSP) were conducted in this study. The title in this study could be inappropriate and hence needs to be changed, since none of results showed the direct evidence showing ability of these two compounds to block “sodium currents”.
  2. Lines 213-214, to monitor intracellular sodium concentration could not indicate sodium influx or intracellular sodium concentration exclusively through voltage-gated sodium channels (Simasko, 1994; PMID: 8166234).
  3. Lines 117-118, the present results tend to be consistent with earlier observations.
  4. Lines 291-296, there is too much speculation regarding the results from Na+ It certainly needs to be emphasized that the magnitude of Na+-Ca2+ exchanging process might be potentially engaged in any changes of intracellular Na+ concentration. The temporal resolution between fluorescent assays and electrophysiological measurements is markedly different. Most of organic compounds indeed can exert multiple actions on Nav, Cav and Kv channels.
  5. Lines 299-301, previous studies have demonstrated the ability of liriodenine to suppress sodium currents.
  6. Line 350, “Dulbecco’s” needs to be corrected.
  7. Figure 3a and lines 143-144, the addition of 80 mM Na+ would have affected the extracellular osmolarity significantly. The tested cells could have been shrunk, and intracellular Na+ content could have been overestimated. Additionally, it needs to be mentioned that voltage-gated Na+ channels should be open by step depolarization. In Figure 3a and Figure 6, none of voltage change was challenged, why could fluorescent signals be elevated over the time course in the scale of seconds? Why could BTX induce such changes with no full elicitation of voltage-gated Na+ channels, although BTX per se is an activator of Na+ current (lines 75-78)?
  8. Perhaps, the investigators need to test the effects of pyrethroid insecticide (e.g., tefluthrin) on intracellular Na+ concentrations (Wu et al., Circulation 2016;134:A15870).

Author Response

Dear reviewer,

 We thank reviewer 1 for their comments and questions, which help us to improve our manuscript.

As asked by the 2nd reviewer, we performed patch-clamp experiments to further characterize the pharmacological activities of five selected alkaloids on hNav1.2, hNav1.3 and hNav1.6 channel subtypes stably expressed in HEK293 cells. The patch-clamp data were very interesting and led to new conclusions and thus the manuscript has been profoundly modified. These experiments were conducted in collaboration and new authors have been added to the authorship.

We have answered to almost all comments and questions (below this letter) and we are pleased to submit this revised manuscript, novely entitled, “Screening an in-House isoquinoline alkaloids library for new blockers of voltage-gated Na+ channels using voltage-sensor fluorescent probes: hits and bias”.

We hope that this revised version responds to all your questions.

best regards,

Christian Legros

Responses to Reviewer 1:

The present study was to explore whether oxostephanine and liriodenine have possible effects on voltage-gated sodium channels in GH3b6 cells. There are several queries regarding the present investigations which are shown below.

 The fluorescent assays with voltage sensor probe (VSP) were conducted in this study. The title in this study could be inappropriate and hence needs to be changed, since none of results showed the direct evidence showing ability of these two compounds to block “sodium currents”.

Our patch-clamp data led to new conclusions and thus we have changed the title for :” Screening an in-House isoquinoline alkaloids library for new blockers of voltage-gated Na+ channels using voltage-sensor fluorescent probes: hits and bias”. We illustrated our new patch-clamp data in 5 novel figures (fig. 7-11).

  1. Lines 213-214, to monitor intracellular sodium concentration could not indicate sodium influx or intracellular sodium concentration exclusively through voltage-gated sodium channels (Simasko, 1994; PMID: 8166234).

We agree with reviewer 1, GH3 cells express other Na+ conducting channels such as NALCN and HCN (Simasko, 1994; PMID: 8166234; Impheng et al., 2021, PMID: 33793981; Chang and Wu, 2021, PMID: 33435511). However, in our conditions, the intracellular increase of Na+ concentration by BTX was totally block by the selective Nav channel inhibitor TTX. To clarify our assay, we added the following sentences in the paragraph “2.3. Effects of oxostephanine, liriodenine, thalmiculine, bebeerine and protopine on Na+ fluxes in a stable cell line expressing hNaV1.3”:

Line 214 : “At 3 µM, BTX induced an increase of ANG-2 fluorescence ratio, which was totally blocked by 1 µM TTX (Fig. 6). This indicates, that BTX only activated hNaV1.3 channels in these conditions. Then, Oxostephanine and liriodenine, efficiently inhibited BTX-induced [Na+]i elevation in HEK293-hNaV1.3 up to 84.34 ± 5.97 % (p < 0.001) and 78.57 ± 11.54 % (p < 0.05) (Fig. 6a,c).”

  1. Lines 117-118, the present results tend to be consistent with earlier observations.

Thanks for this proposition. We added changed the sentence line 115-118 as followed. Our data indicate that GH3b6 cells mainly expressed the neuronal TTX-sensitive NaV1.3 channel subtype and thereby tend to be consistent with earlier observations [35,36,37].

  1. Lines 291-296, there is too much speculation regarding the results from Na+ It certainly needs to be emphasized that the magnitude of Na+-Ca2+ exchanging process might be potentially engaged in any changes of intracellular Na+ concentration. The temporal resolution between fluorescent assays and electrophysiological measurements is markedly different. Most of organic compounds indeed can exert multiple actions on Nav, Cav and Kv channels.

We changed the corresponding paragraph as followed.

Using this Na+ probe, we found that only oxostephanine and liriodenine were able to efficiently inhibit BTX-induced Na+ influx in hNav1.3-HEK293 cells. Indeed, at 10 µM, these two IA almost abolished BTX-induced Na+ influx, suggesting high affinities of both IA for hNav1.3 channels. However, in our assay, thalmiculine, bebeerine and protopine, exhibited modest effects at this concentration. Several hypothesis could be made: 1) thalmiculine, bebeerine and protopine have probably weaker effects on hNav1.3 channels than oxostephanine and liriodenine, 2) thalmiculine, bebeerine and protopine could block other Nav channel subtypes, 3) the spatio-temporal resolution of our assay did not allow to highlight their blocking effects on hNav1.3 channels and 4) these three IA target other ion channels, such as Cav and Kv channels, alleviating any change of [Na+]i by activation of NCX and other transporters. These data also showed the interests but also the limits of using fluorescent probes for screening for Nav channel ligands

  1. Lines 299-301, previous studies have demonstrated the ability of liriodenine to suppress sodium currents.

As asked, we changed this sentence.

“Previous studies have demonstrated the ability of liriodenine to suppress Na+ currents recorded in isolated cardiomyocytes, with an IC50 value in the micromolar range [16,49], which is close to what we observed.”

  1. Line 350, “Dulbecco’s” needs to be corrected.

We are not sure to understand what the reviewer suggested. The commercial name of this product is : Dubelcco’s Modified Eagle Medium

  1. Figure 3a and lines 143-144, the addition of 80 mM Na+ would have affected the extracellular osmolarity significantly. The tested cells could have been shrunk, and intracellular Na+ content could have been overestimated.

Thank you for pointing this imprecision. The final concentration of Na+ is as you noticed 80 mM. But effectively, 100 µL of a depolarizing buffer (160 mM NaCl) was automatically injected to the 100µl incubation buffer in which the cells were incubated (Na-free buffer, VSP buffer). This did not change osmolarity, which were measured at around 300±5  mOsmol before and after the addition of 160 mM NaCl. The TEA concentration was halved (80), and the Na+ went from 0 up to 80 mM.

We added the following sentence in the corresponding section to precise this point.

“The osmolarity of the solution remained the same at around 300±5 mOsmol before and after the addition of the depolarizing solution.”

Additionally, it needs to be mentioned that voltage-gated Na+ channels should be open by step depolarization. In Figure 3a and Figure 6, none of voltage change was challenged, why could fluorescent signals be elevated over the time course in the scale of seconds?  Why could BTX induce such changes with no full elicitation of voltage-gated Na+ channels, although BTX per se is an activator of Na+ current (lines 75-78)?

Toxins, such as BTX or VTD, which inhibit inactivation process and/or induce positive shift of voltage-dependancy of activation, are known to trigger membrane depolarization of excitable cells without any step depolarization (see the review of Wang and Wang doi:10.1016/S0898-6568(02)00085-2). Narahashi has shown (1971, J Physiol., 58:54-70) that BTX induces a progressive and slow depolarization of the nerve membrane only in the presence of physiological Na+ concentration. Quandt and Narahashi (1982, PNAS, 79:6732-36), have shown that BTX induces an increase of Nav channel open probability either at negative potentials. This explain why the addition of such toxins induce slow kinetcs of the fluorescence emission of voltage sensor probe (see  Zhao et al., 2016 Marine Drugs https://doi.org/10.3390/md14020036) or of Na+ probes (Iamshanova et al., 2016, doi:10.1007/s00249-016-1173-7).

In our case, the VSP buffer in Na+ free, so in this condition, BTX bind to Nav channels, but without extracellular Na+, nothing appears, until the injection of 160 mM NaCl solution. Thus, Na+ influx through opend Nav channels, induce membrane depolarization and in turn VSP FRET signal.  Altogether, this results in leaky cells to extracellular Na+. However, the Na+/K+ ATPase is still working for expulsing intracellular Na+. This could explain that ANG-2 and VSP signals slowly reach a plateau, corresponding to an equilibrium between Na+ influx through Nav channels and Na+ efflux mediated by Na+/K+ ATPase and other Na+ transporters (NCX...).

In addition, the VSP probes used here have a high sensitivity, but a low temporal resolution, with an average 500 ms required for the Disbac2,3 used in our assay to migrate inside the cell membrane (see the probes properties from manufacturer, https://www.thermofisher.com/us/en/home/industrial/pharma-biopharma/drug-discovery-development/target-and-lead-identification-and-validation/ion-channel-biology/icb-misc/voltage-sensor-probes.html).

We added lines starting from line 135:

Indeed, BTX binds to open state of Nav channels and consecutively prevents their closing in an irreversibly manner. Then, the injection of Na+ which depolarized the membrane led to the activation of Nav channels and allowed the BTX binding. Since this interaction is irreversible, the cells became leaky to Na+. This led to Na+ influx which was counteracted by Na+ efflux mediated by Na+/K+ ATPase and other Na+ transporters (e.g. NCX). This explain the slow kinetic of fluorescence emission, reaching a plateau, corresponding to an equilibrium between Na+ influx and Na+ efflux.

  1. Perhaps, the investigators need to test the effects of pyrethroid insecticide (e.g., tefluthrin) on intracellular Na+ concentrations (Wu et al., Circulation 2016;134:A15870).

We thank the reviewer for this suggestion. We have already tested pyrethrine to activate Nav channels in GH3b6 cells, but BTX induced VSP signal was better for screening purposes.

Reviewer 2 Report

This manuscript reported the characterization of an antagonist of NaV channels from plants-derived natural products using the voltage sensor probes and BTX measuring the activity of NaVs. My opinion is that the data are clearly presented, the writing and the quality of this paper is generally good. However, as the identified compound oxostephanine shows rather high structure similarity to two previously identified NaVs antagonists, liriodenine and crebanine, the author should justify the novelty of oxostephanine as a novel NaV antagonist. Furthermore, liriodenine and crebanine only partially inhibit the VSP signal in GH3b6 cells which express a mix of NaV subtypes (NaV1.2, NaV1.3 and NaV1.6), and they almost do not affect the ANG-2 signal when testing on the cloned NaV1.3 channel, the authors are encouraged to test the subtype selectivity of these three compounds on NaV1.2, NaV1.3 and NaV1.6 channels to more explicitly illustrate their findings.

Abstract:

1) As oxostephanine, liriodenine, and crebanine all share a similar structure scaffold, and the latter two compounds have been reported to be NaVs antagonist, the author should justify how does this research advance our understanding of NaV blockers (what’s the novelty of oxostephanine as NaV antagonist)?

Introduction:

1) Page 2, lines 51-52: NaV inhibitors are also used for the treatment of epilepsy, such as carbamazepine, so NaV antagonists as anti-epilepsy drugs (AED) should be mentioned here.

2) Page 2, lines 64-65: Currently, there are 8 identified neurotoxin binding sites on NaVs (sites 1-6, plus the ICA/PF site and the LA site), not 6 sites (Ref: Voltage-Gated Sodium Channels: Structure, Function, Pharmacology, and Clinical Indications; doi: 10.1021/jm501981g).

3) Fig. 1: the sentence ‘two microbial alkaloids with guanidinium group block of Nav channels’ should be re-written (Page 3, Line 96).

Results:

1)the meaning of the subsection title ‘GH3b6 cells are suitable for pharmacological characterization of NaV channels’ is unclear and should be rewritten (Page 4, line 107).  

2) Page 4, line 110: there is only Fig. 2, no Fig. 2A.

3) Page 4, line 120: the sentence ‘GH3b6 cells express TTX-S Nav channels expressed in GH3b6 cells’ should be rewritten.

4) Page 4, lines 137-138:I think what the author want to express is that GH3b6 cells is suitable for FRET-based VSP assay for screening NaV inhibitors, then this sentence should be rewritten.

5) Fig. 3a: It’s unclear whether these are representative traces from the same single well (repetitive measurement of the fluorescence of a single well of cells by sequentially adding different doses of BTX), or from different wells of cells using different doses of BTX treatment (different groups). From the description in Materials and methods, it might be the latter. Then, if from different wells, does every trace represent the MEAN data from several different testing wells of the same BTX dose? What’s the filled circle on each curve means (showing the timepoint of sampling?)?  For the reader’s convenience, the author should make these details clear.

6) legend for Fig. 3b and 3c: In Fig. 3b, it seems that the curve was constructed by plotting the normalized BTX response (normalizing the response at each BTX dose to that of 3 µM BTX) as a function of the BTX concentration. In Fig. 3c, did the author use 1 µM BTX to activate the channel and normalize the response of each dose of TTX to that of control (1 µM BTX)? If it’s the case, then the legend caused ambiguity and should be rewritten.

The author stated ‘Data are presented as mean ± SEM, calculated from at least n = 3 replicates and representative of at least 2 independent experiments’ in the Data analysis subsection of ‘Materials and methods’, then did the data on the curve represent mean±sem values from at least 6 testing wells?

7) Fig. 4a: As I mentioned before, is each testing from the same well of cells (does blue and red curves represent two independent groups?)?  The unit of the x-axis is second or minute?  As in Fig. 3a, it says the unit of the time axis is second, but the legend here says the unit of x-axis is minute.

8) the n value is missing in Fig. 5.

9) Page 8, line 215 and line 219: Fig. 6c does not show any data for oxostephanine and liriodenine, and Fig. 6b does not show any data for IA69, IA52 and IA49.

10) In Fig. 5b, IA49, IA52, and IA69 partially inhibited the VSP signal, which actually resembles the action of some gating modifier toxins on NaVs and KVs (drug-bound channels could be re-opened with the drug as a cargo). However, in Fig. 6C, these three compounds almost do not affect the Na+ inflow. The VSP and ANG-2 tests are basically the same as for monitoring the activity of NaV channels, the difference between Fig. 5 and Fig. 6 is that the author used transfected NaV1.3 in Fig. 6 but mixed NaV channels (NaV1.3, NaV1.2, and NaV1.6) in Fig. 5, therefore, one reasonable explanation is that these three compounds could inhibit NaV1.2 and NaV1.6 but not NaV1.3 (or weak inhibition on NaV1.3). The authors are encouraged to add some new data to illustrate this possibility (patch-clamp test is preferred).

Discussion:

1) Page 11, lines 277-278, and lines 291-296: As I mentioned before, another possibility is that bebeerine, thalmiculine, and protopine inhibit NaV1.2 and/or NaV1.6 but not NaV1.3.

Methods:

1) Page 14, lines 440-443: I’m getting confused with the description of data analysis here, could the author make it much clearer? As well as the data analysis method described on Page 15, Lines 450-452.  What’s the sampling frequency in Fig. 4a?

Author Response

Dear reviewer,

 We thank reviewer 2 for all her/his comments and questions, which help us to improve our manuscript.

As requested, we performed patch-clamp experiments to further characterize the pharmacological activities of five selected alkaloids on hNav1.2, hNav1.3 and hNav1.6 channel subtypes stably expressed in HEK293 cells. The patch-clamp data were very interesting and led to new conclusions and thus the manuscript has been profoundly modified. These experiments were conducted in collaboration and new authors have been added to the authorship.

We have answered to almost all comments and questions (below this letter) and we are pleased to submit this revised manuscript, novely entitled, “Screening an in-House isoquinoline alkaloids library for new blockers of voltage-gated Na+ channels using voltage-sensor fluorescent probes: hits and bias”.

We hope that this revised version responds to all your questions.

best regards,

Christian Legros

Responses to Reviewer 2:

This manuscript reported the characterization of an antagonist of NaV channels from plants-derived natural products using the voltage sensor probes and BTX measuring the activity of NaVs. My opinion is that the data are clearly presented, the writing and the quality of this paper is generally good. However, as the identified compound oxostephanine shows rather high structure similarity to two previously identified NaVs antagonists, liriodenine and crebanine, the author should justify the novelty of oxostephanine as a novel NaV antagonist.

Furthermore, liriodenine and crebanine only partially inhibit the VSP signal in GH3b6 cells which express a mix of NaV subtypes (NaV1.2, NaV1.3 and NaV1.6), and they almost do not affect the ANG-2 signal when testing on the cloned NaV1.3 channel, the authors are encouraged to test the subtype selectivity of these three compounds on NaV1.2, NaV1.3 and NaV1.6 channels to more explicitly illustrate their findings.

Answer:

Liriodenine and oxostephanine are two oxoaporphines, while crebanine is an aporphine. It is right that liriodenine and crebanine shares about 80% of structure similarity, while liriodenine and oxostephanine shares 95.3% of structure similarity. Thereby, in our study, we did not use crebanine. In VSP and ANG2 assays, the blocking effects of liriodenine and oxostephanine almost reached 100%. In VSP assays, the inhibition induced by liriodenine reached 99% and 78% in the ANG2 assay. For oxostephanine, the inhibition rate reached 80-84% in both assays. However, we agree with reviewer 2 that crebanine like liriodenine have been proven to exhibit anti-arrythmic activities mediated by their blocking action on Nav, Kv and Cav channels. The structural similarity likely explain these common activities.

As requested by the reviewer, we performed patch-clamp experiments on three cell lines, expressing hNav1.2, hNav1.3 and hNav1.6 channels. We obtained interesting data illustrated in figures 7-11 (and one supplementary figure S7). Due to the lack of solubility of the five isoquinoline alkaloids (IA) in aqueous solution and non specific effects of DMSO in patch-clamp recordings, we could not test them at concentration above to 10 µM.

These data showed that :

  • Only bebeerine (10 µM) exhibited a strong inhibitory effects on hNav1.2 and hNav1.6 channels and a weak effect on hNav1.3 channels
  • Liriodenine (10 µM) exhibited a weak inhibitory effect on Na+ currents elicited by hNaV1.3 channels after 1 Hz depolarizing pulse, revealing an use-dependent blockade mechanism.
  • At 10 10 µM each, oxostephanine and liriodenine decreased the effects of BTX on the activation and inactivation properties of hNav1.3 channels, indicating that they bind to open channels and all better in the presence of BTX.

It is possible, that oxostephanine, liriodenine, thalmiculine and protopine could inhibit hNav1.2, hNav1.3 and hNav1.6 channels at higher concentrations. In addition, oxostephanine could also exhibit as liriodenine a use-dependent mechanism of block of hNav1.3 at higher concentrations.

Abstract:

1) As oxostephanine, liriodenine, and crebanine all share a similar structure scaffold, and the latter two compounds have been reported to be NaVs antagonist, the author should justify how does this research advance our understanding of NaV blockers (what’s the novelty of oxostephanine as NaV antagonist)?

Anwer: the abstract was changed according the patch-clamp data.

Introduction:

1) Page 2, lines 51-52: NaV inhibitors are also used for the treatment of epilepsy, such as carbamazepine, so NaV antagonists as anti-epilepsy drugs (AED) should be mentioned here.

Anwer: Yes course, we added this category of drugs which inhibit Nav channels.

2) Page 2, lines 64-65: Currently, there are 8 identified neurotoxin binding sites on NaVs (sites 1-6, plus the ICA/PF site and the LA site), not 6 sites (Ref: Voltage-Gated Sodium Channels: Structure, Function, Pharmacology, and Clinical Indications; doi: 10.1021/jm501981g).

Anwer: We corrected this error.

3) Fig. 1: the sentence ‘two microbial alkaloids with guanidinium group block of Nav channels’ should be re-written (Page 3, Line 96).

Anwer: We changed this sentence as followed:

Tetrodotoxin and saxitoxin are two marine alkaloids with guanidinium group which block of Nav channels.

Results:

1)the meaning of the subsection title ‘GH3b6 cells are suitable for pharmacological characterization of NaV channels’ is unclear and should be rewritten (Page 4, line 107). 

Anwer:

Thank you for pointing out the approximation, the sentence was reformulated.

“2.1 VSP-FRET assays using GH3b6 for pharmacological characterization of Nav channels”

2) Page 4, line 110: there is only Fig. 2, no Fig. 2A.

Anwer:

This typo was corrected in L111. Thank you for pointing it out.

3) Page 4, line 120: the sentence ‘GH3b6 cells express TTX-S Nav channels expressed in GH3b6 cells’ should be rewritten.

Anwer:

The title of fig.2 was reformulated (Figure 2. GH3b6 cells express TTX-S Nav channels.).

4) Page 4, lines 137-138:I think what the author want to express is that GH3b6 cells is suitable for FRET-based VSP assay for screening NaV inhibitors, then this sentence should be rewritten.

Anwer:

You are indeed assuming correctly; the sentence was reformulated to reflect this idea more clearly. “Altogether, these results demonstrate that GH3b6 cells are suitable for FRET-based VSP assay for screening Nav channels inhibitors.”

5) Fig. 3a: It’s unclear whether these are representative traces from the same single well (repetitive measurement of the fluorescence of a single well of cells by sequentially adding different doses of BTX), or from different wells of cells using different doses of BTX treatment (different groups). From the description in Materials and methods, it might be the latter. Then, if from different wells, does every trace represent the MEAN data from several different testing wells of the same BTX dose? What’s the filled circle on each curve means (showing the timepoint of sampling?)?  For the reader’s convenience, the author should make these details clear.

Anwer:

Fig. 3a represents as you correctly guessed, each trace is representative from three wells with a given BTX concentration. We surimposed all traces obtained with different concentrations of BTX. Error bars were omitted for clarity. Each filled circle represents one timepoint sampling (sampling was done sequentially on each well of a 96well plate). Clarification of the legend was made in L145 to L148.

 “Representative kinetics traces illustrating the fluorescence emission of CC2-DMPE (λ = 460nm, blue line) and DisBAC2(3) (λ = 580 nm, red line) in the presence of increasing concentrations of BTX (0.01 to 3 µM) before and after raising Na+ concentration to 80 mM. Data are mean of three wells.”

6) legend for Fig. 3b and 3c: In Fig. 3b, it seems that the curve was constructed by plotting the normalized BTX response (normalizing the response at each BTX dose to that of 3 µM BTX) as a function of the BTX concentration. In Fig. 3c, did the author use 1 µM BTX to activate the channel and normalize the response of each dose of TTX to that of control (1 µM BTX)? If it’s the case, then the legend caused ambiguity and should be rewritten.

Anwer: Legends b and c were modified to avoid the guessing part from the reader. Thank you for noticing.

The author stated ‘Data are presented as mean ± SEM, calculated from at least n = 3 replicates and representative of at least 2 independent experiments’ in the Data analysis subsection of ‘Materials and methods’, then did the data on the curve represent mean±sem values from at least 6 testing wells?

Anwer: We clarified this in the DATA analysis section. For VSP experiments showed in figure 3b,c, data are mean ± SEM (n=3 wells) from one representative experiment. This concentration-response was done twice and data reported in Table 2 is the mean of two experiments. For ANG-2 assay, data showed in 6b are mean ± SEM of three independent experiments. We corrected the caption and the material and method section, according to these comments. Each point showed in figure 6a and c, corresponds to the mean of 3 wells from one plate.

7) Fig. 4a: As I mentioned before, is each testing from the same well of cells (does blue and red curves represent two independent groups?)?  The unit of the x-axis is second or minute?  As in Fig. 3a, it says the unit of the time axis is second, but the legend here says the unit of x-axis is minute.

Anwer: The blue and red traces were obtained from distinct experiments. We added precision in the corresponding caption. We corrected the indication of the horizontal axis title: Time (s).

8) the n value is missing in Fig. 5.

Anwer: Thank you for noticing imprecisions in the legend. This has been corrected.

9) Page 8, line 215 and line 219: Fig. 6c does not show any data for oxostephanine and liriodenine, and Fig. 6b does not show any data for IA69, IA52 and IA49.

Anwer: Correction in the text were made (lines 235, 238).

10) In Fig. 5b, IA49, IA52, and IA69 partially inhibited the VSP signal, which actually resembles the action of some gating modifier toxins on NaVs and KVs (drug-bound channels could be re-opened with the drug as a cargo). However, in Fig. 6C, these three compounds almost do not affect the Na+ inflow. The VSP and ANG-2 tests are basically the same as for monitoring the activity of NaV channels, the difference between Fig. 5 and Fig. 6 is that the author used transfected NaV1.3 in Fig. 6 but mixed NaV channels (NaV1.3, NaV1.2, and NaV1.6) in Fig. 5, therefore, one reasonable explanation is that these three compounds could inhibit NaV1.2 and NaV1.6 but not NaV1.3 (or weak inhibition on NaV1.3). The authors are encouraged to add some new data to illustrate this possibility (patch-clamp test is preferred).

Anwer: Data have been obtained and now described in the revised manuscript.

Methods:

  • Page 14, lines 440-443: I’m getting confused with the description of data analysis here, could the author make it much clearer? As well as the data analysis method described on Page 15, Lines 450-452.

What’s the sampling frequency in Fig. 4a?

Answer:

We corrected this as described above. Sampling frequency was also indicated in material and method section (4.4. Fluorescent assays). The plate reader measured fluorescence emission each 5 s.

.

Round 2

Reviewer 1 Report

The authors have answered most of questions raised. Acceptable!

Author Response

thanks for your peer-reviewing.

Reviewer 2 Report

The authors added some new data to address the activity of the IA compounds screened out by VSP technology, which showed that the VSP strategy might not be reliable for identifying NaV antagonists. Among the five hit IA compounds (liriodenine, oxostephanine, thalmiculine, protopine, and bebeerine) in VSP experiments, only bebeerine was proved to be an effective NaV1.2 and NaV1.6 inhibitor as determined by the gold standard patch-clamp method. This study includes interesting data in addressing the shortcomings of the VSP screening method, however, the authors focus too much on the initial screening results as they used lots of experiments to address why liriodenine and oxostephanine are not effective in inhibiting NaVs but spared the actual NaV antagonist bebeerine. Moreover, the patch-clamp experiments in Fig. 10 and Fig. 11 are not properly designed and can not draw the conclusion that liriodenine and oxostephanine binds to the opening NaV channels.  In my opinion, the authors should focus on the hit compound bebeerine to study its activity and action mechanism. Therefore, this manuscript needs another major revision.

Abstract: Lines 38-39: what’s the 1 Hz stimulation protocol? What’s the depolarization time?

Abstract: Lines 40-41: As shown in Figure 8, liriodenine BUT NOT oxostephanine decreased the V1/2 shift caused by BTX. In Figure 9, oxostephanine but not liriodenine changed the inactivation of NaV1.3 channel. As judged from the current traces, the acceleration of inactivation by IA14 is not evident.

Fig 3B legend: this is not FRET signal but inhibition of FRET signal (the fluorescence ratio indicates the decrement of FRET).

Fig 3C: What’s the Y axis?

Fig. 3C legend: The author illustrated the Y axis as ‘BTX-induced FRET signal were blocked by TTX’. It’s not ‘BTX-induced FRET’, at resting membrane potential, the FRET signal was the maximum and BTX depolarized the membrane and destroyed the FRET signal. Therefore, this should be recovery of FRET signal by TTX.

Fig 4a legend: X is time and Y should be the fluorescence ratio.

In Fig 5a-b, the Y axis was stated as ‘VSP signal ratio’, in Figure 4b, the Y axis is ‘inhibition of BTX-induced VSP signal’, in Fig 3b, the Y axis is ‘fluorescence ratio’. Actually, these all are plotting the fluorescence ratio (λ460 nm/ λ 580 nm) or normalized fluorescence ratio as a function of compounds concentration (BTX, TTX, and various IAs), therefore, the authors should make the Y axis consistent.

Line 233-234: the signal is inhibited by TTX, why the authors conclude that only NaV1.3 is activated by BTX here? NaV1.2 and NaV1.6 are also TTX-S NaVs.

Fig. 6b: what is control? Bath solution perfusion? And from Fig 6a, only approximately 50% inhibition was observed.

Fig.7a is not cited in the main text. How was the compound applied? Acute application or pre-incubation? If pre-incubation, how can be the traces put together for comparison as they were not from the same cell (the currents before and after compound treatment).

Fig 9: The unit of AUC should be pA×ms? The author should highlight the -20 mV current trace.

Lines 304-306: from Fig 9c, IA-39 did not significantly change the inactivation rate of NaV1.3.

Fig 9c legend is not related to the figure.

Fig 10 can not give the conclusion that liriodenine inhibit NaV1.3 use-dependently. 1) what’s the use-dependent inhibition voltage protocol, what’s the depolarization time at +10 mV and what’s the recovery time? The author stated the activation curve was built by 2000 ms-lasting depolarizations from -130 mV to +40 mV (sweep interval = 5s), then the frequency is only 1/7 Hz; 2) the frequency of 1 Hz is too low to test use-dependent inhibition as NaV1.3 recovers from fast inactivation very fast (within 100 ms). From Fig 10C, the traces for IA39 and BTX+IA39 superimposed. And why the author uses IA39 and liriodenine randomly (Fig 10c, it should not be BTX+IA39?). The currents traces in Fig 10a seems weird, as BTX, BTX+IA14, and BTX+IA39 did not slow down the fast inactivation of NaV1.3 before 1 Hz stimulation. Theoretically, the fast inactivation of NaV1.3 should be slowed by BTX no matter what the voltage stimulation protocol is, as observed in Fig. 9a.

Fig 11: The author here used the time constant of fast inactivation to measure the inactivation rate of NaV, however, the figures obviously are not showing the inactivation time constant, whose unit should be ‘ms’. Again, as the 1 Hz protocol used for analyzing the use-dependent inhibition in Fig 10 is not suitable, the analysis of tau inactivation does not make sense.

As IA14 and IA39 did not affect the currents of NaV1.2, NaV1.3 and NaV1.6 as determined by the gold standard patch-clamp analysis, I suggest the author to focus on investigating the effect the positive hit, bebeerine, on NaVs.  The data in Fig. 10-11 did not add any useful information to the manuscript, and Fig. 8 and Fig. 9 could be moved to supplementary information.

Author Response

Response to Reviewer 2:

We would like to thank Reviewer #2 for helping us to improve our manuscript, thanks to relevant suggestions. Here are our responses to the different comments, and the updated document.

1-“This study includes interesting data in addressing the shortcomings of the VSP screening method, however, the authors focus too much on the initial screening results as they used lots of experiments to address why liriodenine and oxostephanine are not effective in inhibiting NaVs but spared the actual NaV antagonist bebeerine. Moreover, the patch-clamp experiments in Fig. 10 and Fig. 11 are not properly designed and can not draw the conclusion that liriodenine and oxostephanine binds to the opening NaV channels. In my opinion, the authors should focus on the hit compound bebeerine to study its activity and action mechanism.”

Answer: We agree with this comment! We modified the manuscript as suggested. We transfered our patch-clamp data in the first revised manuscript, concerning oxostephanine and liriodenine to supplementary material section. We further characterized the effects of Bebeerine on hNav1.2, hNav1.3 and hNav1.6 channels by automated patch-clamp electrophysiology. Concerning the mechanism of action of oxostephanine and liriodenine, we modified our conclusions.

“Abstract: Lines 38-39: what’s the 1 Hz stimulation protocol? What’s the depolarization time?”

Answer: This was removed from the abstract. The end of the abstract was changed.

“Abstract: Lines 40-41: As shown in Figure 8, liriodenine BUT NOT oxostephanine decreased the V1/2 shift caused by BTX. In Figure 9, oxostephanine but not liriodenine changed the inactivation of NaV1.3 channel. As judged from the current traces, the acceleration of inactivation by IA14 is not evident.”

Answer: Since we focused on bebeerine, this sentence was removed from the abstract.

“Fig 3B legend: this is not FRET signal but inhibition of FRET signal (the fluorescence ratio indicates the decrement of FRET). Fig 3C: What’s the Y axis?”

Answer: This is right, indeed FRET signal and VSP signal change in opposite direction after depolarization. Less FRET occurs, higher is the fluorescence ratio or VSP signal ratio. The legend was corrected by using VSP signal instead of FRET signal, and as in the first version, the Y title was changed. For clarity, we surimposed BTX-induced responses and TTX-inhibition curves in one chart (Fig. 3b). Y axis is VSP signal ratio. 

“Fig. 3C legend: The author illustrated the Y axis as ‘BTX-induced FRET signal were blocked by TTX’. It’s not ‘BTX-induced FRET’, at resting membrane potential, the FRET signal was the maximum and BTX depolarized the membrane and destroyed the FRET signal. Therefore, this should be recovery of FRET signal by TTX.”

Answer: This is right and as explained above, this was corrected.

“Fig 4a legend: X is time and Y should be the fluorescence ratio.”

Answer: Axis titles were added in this figure.

“In Fig 5a-b, the Y axis was stated as ‘VSP signal ratio’, in Figure 4b, the Y axis is ‘inhibition of BTX-induced VSP signal’, in Fig 3b, the Y axis is ‘fluorescence ratio’. Actually, these all are plotting the fluorescence ratio (λ460 nm/ λ 580 nm) or normalized fluorescence ratio as a function of compounds concentration (BTX, TTX, and various IAs), therefore, the authors should make the Y axis consistent.”

Answer: Thank you for the suggestion, harmonizing the Y labels will add clarity, Y labels on figures 5 a-b, 4 b-c is now VSP signal ratio.

“Line 233-234: the signal is inhibited by TTX, why the authors conclude that only NaV1.3 is activated by BTX here? NaV1.2 and NaV1.6 are also TTX-S NaVs.”

Answer: I am not sure to correcttl localize this, but I think it is in line 155-156. We removed “such as Nav1.3 GH3b6 plasma membrane”, since we have no argument here to conclude that. However, if it concerns ne the paragraph 2.3, the following sentence “This indicates, that BTX only activated hNav1.3 channels in these conditions.” We removed it, since it is not relevant here.

“Fig. 6b: what is control? Bath solution perfusion? And from Fig 6a, only approximately 50% inhibition was observed.”

Answer: ANG-2 was used as Fura-2, we described the procedure in section 4.4. As control, HBSS was injected or HBSS containing 1 µM TTX, both traces matched most of the time. We changed the example of kinetic trace in Fig. 6a. Indeed, the old one was not representative. This figure was revised, since we repeated the experiments. We added and histogram to illustrate the weak inhibition effect of protopine, thalmiculine and bebeerine on ANG-2 signal induced by BTX. With more data, it appears that Bebeerine had an inhibitory effect on BTX-induced ANG-2 signal, but weaker than oxostephanine and liriodenine.

“Fig.7a is not cited in the main text. How was the compound applied? Acute application or pre-incubation? If pre-incubation, how can be the traces put together for comparison as they were not from the same cell (the currents before and after compound treatment).”

Answer: The previous figure 7 was replaced by a new one in which we added new data fo bebeerine. Fig. 7a, illustrating current traces was then cited in the main text. Sorry, we forgot to indicate the incubation time, we added it in the material and method section. “Each IA was preincubated 10 min before recordings”. Fig.7a was changed, since the illustrative traces are from different wells.

“Fig 9: The unit of AUC should be pA×ms? “

Answer: That’s right. Thus, we corrected this error.

“ The author should highlight the -20 mV current trace.

Lines 304-306: from Fig 9c, IA-39 did not significantly change the inactivation rate of NaV1.3.

Fig 9c legend is not related to the figure.”

Answer : We analysed new patch-clamp data to better characterize the effects of oxostephanine and liriodenine in the presence of BTX on Nav1.3  channels. Then, we changed figure 9 and we only showed the effects of IA14 and IA39 on current densities, V1/2 activation and AUC in the presence of BTX. In Fig. 9a, the -60 mV and -20 mV were highlighted to illustrate the strong effects of BTX on both activation and inactivation properties of Nav channels.

“Fig 10 can not give the conclusion that liriodenine inhibit NaV1.3 use-dependently. 1) what’s the use-dependent inhibition voltage protocol, what’s the depolarization time at +10 mV and what’s the recovery time? The author stated the activation curve was built by 2000 ms-lasting depolarizations from -130 mV to +40 mV (sweep interval = 5s), then the frequency is only 1/7 Hz; 2) the frequency of 1 Hz is too low to test use-dependent inhibition as NaV1.3 recovers from fast inactivation very fast (within 100 ms). From Fig 10C, the traces for IA39 and BTX+IA39 superimposed. And why the author uses IA39 and liriodenine randomly (Fig 10c, it should not be BTX+IA39?). The currents traces in Fig 10a seems weird, as BTX, BTX+IA14, and BTX+IA39 did not slow down the fast inactivation of NaV1.3 before 1 Hz stimulation. Theoretically, the fast inactivation of NaV1.3 should be slowed by BTX no matter what the voltage stimulation protocol is, as observed in Fig. 9a.”

“Fig 11: The author here used the time constant of fast inactivation to measure the inactivation rate of NaV, however, the figures obviously are not showing the inactivation time constant, whose unit should be ‘ms’. Again, as the 1 Hz protocol used for analyzing the use-dependent inhibition in Fig 10 is not suitable, the analysis of tau inactivation does not make sense.”

Answer : Thanks for this comment, we removed these data from the manuscript.

“As IA14 and IA39 did not affect the currents of NaV1.2, NaV1.3 and NaV1.6 as determined by the gold standard patch-clamp analysis, I suggest the author to focus on investigating the effect the positive hit, bebeerine, on NaVs.  The data in Fig. 10-11 did not add any useful information to the manuscript, and Fig. 8 and Fig. 9 could be moved to supplementary information.”

Answer : Thank you for all comments. As asked, we removed figures 10 and 11 and transfered previous Fig. 8 and Fig. 9 to  supplementary information. We also removed figure 12, since we have no effects of oxostephanine and liriodenine in patch-clamp experiments.